# Basal friction of Fleming Glacier, Antarctica, Part A: sensitivity of inversion to temperature and bedrock uncertainty

Chen Zhao[1,3], Rupert M. Gladstone[2], Roland C. Warner[3], Matt A. King[1], Thomas Zwinger[4] , Mathieu Morlighem[5]

[1] School of Technology, Environments and Design, University of Tasmania, Hobart, Australia
[2] Arctic Centre, University of Lapland, Rovaniemi, Finland
[3] Antarctic Climate and Ecosystems Cooperative Research Centre, University of Tasmania, Hobart, Australia
[4] CSC-IT Center for Science Ltd., Espoo, Finland
[5] Department of Earth System Science, University of California Irvine, CA 92697-3100, USA

**Abstract**

Many glaciers in the Antarctic Peninsula are now rapidly losing mass. Understanding of the dynamics of these fast-flowing glaciers, and their potential future behavior, can be improved through ice sheet modeling studies. Inverse methods are commonly used in ice sheet models to infer the spatial distribution of a basal friction coefficient, which has a large effect on the basal velocity and ice deformation. Here we use the full-Stokes Elmer/Ice model to simulate the Wordie Ice Shelf-Fleming Glacier system in the southern Antarctic Peninsula. With an inverse method, we infer the pattern of the basal friction coefficient from surface velocities observed in 2008. We propose a multi-cycle spin-up scheme to reduce the influence of the assumed initial englacial temperature field on the final inversion. This is particularly important for glaciers like the Fleming Glacier, which have areas of strongly temperature-dependent, deformational flow in the fast-flowing regions. Sensitivity tests using various bed elevation datasets, ice front positions and boundary conditions demonstrate the importance of high-accuracy ice thickness/bed geometry data and precise location of the ice front boundary.

## 1 Introduction

In response to rapid changes in both atmosphere and ocean, glaciers in West Antarctica (WA) and the Antarctic Peninsula (AP) have undergone rapid dynamic thinning and increased ice discharge over recent decades, which has led to a significant contribution to global sea level rise (Cook et al., 2016; Gardner et al., 2018; Wouters et al., 2015). Understanding the underlying processes is crucial to improve modeling of ice dynamics and enable reliable predictions of contributions to sea level change, especially for fast-flowing outlet glaciers.

The high velocities of fast-flowing outlet glaciers arise from internal ice deformation or ice sliding at the bed or both. Internal deformation is dependent on gravitational driving stress, englacial temperature, the development of anisotropic structure at the grain scale in polycrystalline ice (e.g. Gagliardini et al. (2009)) and larger scale weakening from fractures (Borstad et al., 2013). Basal sliding is dependent on the gravitational driving stress, bedrock topography and the basal slipperiness, which in turn is affected by the roughness of the bed, the presence of deformable till, or subglacial hydrology. Therefore, one of the keys to modeling fast-flowing glaciers is accurate knowledge of the basal conditions: the bedrock topography and the basal slipperiness (Gillet-Chaulet et al., 2016; Schäfer et al., 2012). Inverse methods are commonly used in ice sheet models to infer the basal friction coefficient,

basal velocities, and ice rheology from the glacier geometry and observed surface velocities (Gillet-Chaulet et al., 2016; Gladstone et al., 2014; Morlighem et al., 2010).

Poorly constrained quantities, like basal topography, and the distribution of internal temperature, have provided major challenges for modeling the basal shear stress (Vaughan and Arthern, 2007). However, in studies carried out on a fast-flowing outlet glacier draining

from the Vestfonna ice cap in the Arctic (Schäfer et al., 2014; Schäfer et al., 2012), it was found that the Robin inverse method did not depend strongly on the uncertainties in the topographic and velocity data. In their case, sliding dominated the flow regime, and the impact of internal deformation on ice velocity was relatively small compared to the important role of friction heating at the bed on the basal sliding (Schäfer et al., 2014; Schäfer et al.,

2012). It is unclear whether this property is specific to the Vestfonna situation or if it also applies to other fast flowing glaciers. The motivation of this paper is twofold: to test the sensitivity of a variational inverse method (MacAyeal, 1993; Morlighem et al., 2010) for basal friction to basal geometry and to an assumed initial englacial temperature distribution for a different outlet glacier system, and to determine a robust basal friction coefficient

pattern for the Fleming Glacier, located in the southern AP, in 2008.

The Wordie Ice Shelf (WIS) (Fig. 1b) in the southern AP has experienced ongoing retreat and collapse since 1966, with its almost-complete disappearance by 2008 (Cook and Vaughan, 2010; Zhao et al., 2017). The Fleming Glacier (FG) (Fig. 1b), the main tributary glacier that fed the WIS, has a current length of ~80 km and is ~10 km wide near the ice front (Friedl et

al., 2018). This glacier has recently shown a rapid increase in surface-lowering rates (doubling near the ice front after 2008) (Zhao et al., 2017), and the largest velocity changes (> 500 m yr$^{-1}$ near the ice front) across the whole Antarctic ice sheet over 2008-2015 (Walker and Gardner, 2017).

In this study, we employ the Elmer/Ice code (Gagliardini et al., 2013), a three-dimensional

(3D), finite element, full-Stokes ice sheet model, to invert for the basal friction coefficient distribution over the whole WIS-FG system using a parallel computing environment. We assess its sensitivity to assumptions about the initial temperature distribution, bedrock topographies, ocean boundary conditions and other parameters in the model. We introduce the data in Sect. 2, present the ice sheet model, spin-up scheme and experiment design in Sect. 3,

and discuss the results in Sect. 4 before we give the conclusions in Sect. 5.

**2 Data**

**2.1 Surface elevation data in 2008**

The surface topography in 2008 (Fig. 2a) is combined from two SPOT DEM products acquired on 21$^{st}$ Feb, 2007 (resolution: 240 m) and 10$^{th}$ Jan, 2008 (resolution: 40 m) (Korona

et al., 2009) and an ASTER DEM product ranging from 2000 to 2009 (resolution: 100m) (Cook et al., 2012). The surface elevation data for the Fleming Glacier is mainly from the SPOT DEM product acquired on 10$^{th}$ Jan, 2008 (see masks of different DEM products in Fig. S1 in the supplementary material). Here we apply the SPOT DEM precision quality masks on the raw data to extract the DEM data with correlation scores from 20% to 100%. Areas with

low correlation scores were filled with the ASTER DEM data. To remove noise from the DEM data, the combined DEM (resolution: 40 m) is resampled to 400 m with a median filter and a window size of 10×10 pixels.

Both SPOT and ASTER DEM products used the EGM96 geoid (Lemoine et al., 1998) as the height reference. However, the bed elevation data from Bedmap2 dataset (Fretwell et al.,

2013) adopted the EIGEN-GL04C geoid (Förste et al., 2008) as its height reference, and we chose to convert all the elevation datasets to the WGS84 ellipsoid. The EGM96 geoid (Lemoine et al., 1998) and EIGEN-GL04C geoid (Förste et al., 2008) are used to convert from the EGM96 geoid and EIGEN-GL04C geoid values to WGS84 ellipsoidal heights,

respectively. We extract a median value of 15 m for the DEM data over Marguerite Bay (Fig. 1a) as the mean local sea level in the ellipsoid frame.

Both geoid-ellipsoid separation fields vary very slowly spatially compared to the surface elevation of the ice sheet, so that we do not expect any significant change in the computed surface slope that enters the driving stress calculations from mapping the geoid-based elevations into the ellipsoidal frame. Ice thickness is preserved in converting the datasets to the ellipsoid reference frame (see Sect. 2.2). Clearly, the sea level height in the ellipsoidal reference frame enters the calculation of ocean water pressure on the ice front explicitly, as we discuss under experimental design in Sect 3.6 and Sect. 4.4.

## 2.2 Bed elevation data

The bed topography plays an important role in the basal sliding and distribution of fast-flowing ice (De Rydt et al., 2013). However, high-resolution observations of bedrock elevation for the WIS-FG system are still not available. To explore the sensitivity of the basal friction coefficient distribution to the uncertainty in the bedrock topography, we adopt three basal topographies. The first is from the Bedmap2 dataset (Fretwell et al., 2013) with a resolution of 1 km (hereafter bed_bm; Fig. 2b), which we converted from the EIGEN-GL04C geoid (Förste et al., 2008) to WGS84 ellipsoid heights. The other two are derived using the equations below:

$$\text{bed\_zc} = S_{2008} - H_{mc} \tag{1}$$

$$\text{bed\_mc} = S_{bm} - H_{mc} \tag{2}$$

where $S_{2008}$ is the 2008 surface DEM described in Sec. 2.1, and $S_{bm}$ is the surface elevation data from Bedmap2 (Fretwell et al., 2013), again relative to the WGS84 ellipsoid. $S_{bm}$ is downscaled to 500 m with a bilinear interpolation method. $H_{mc}$ (where "mc" refers to "mass conservation") is the ice thickness data with a resolution of 450 m covering three regions shown in Fig. 2e. $H_{mc}$ for the yellow area is computed using the Ice Sheet System Model's mass conservation method (Morlighem et al., 2011; Morlighem et al., 2013), based on ice thickness measurements from the Center for Remote Sensing of Ice Sheets (CReSIS), using ice surface velocities in 2008 from Rignot et al. (2011b), surface accumulation from RACMO 2.3 (van Wessem et al., 2016) and 2002-2008 ice thinning rates from Zhao et al. (2017). The thickness data for the grey area is interpolated from Bedmap2 (Fretwell et al., 2013), while the data in the red area ensures a smooth transition between the two regions. The yellow area indicates the Fleming Glacier system with ice velocity >100 m yr[-1]. The uncertainty of $H_{mc}$ (Fig. 2f) ranges from 10 m to 108 m. For the calculation of $H_{mc}$, we assume that the ice elevation changes over 2002 to 2008 (Zhao et al., 2017) were small compared to the uncertainties in ice thickness (Fig. 2f) and could be ignored in the ice thickness measurements which span a wider time frame. Both bed_mc (Fig. 2c) and bed_zc (Fig. 2d) have a higher resolution of 450 m while bed_bm (Fig. 2b) has a resolution of 1 km. The uncertainty of bed_bm for the fast-flowing regions of the Fleming Glacier (yellow and red area in Fig. 2e) ranges from 151 m to 322 m (Fretwell et al., 2013), while the uncertainty of bed_mc and bed_zc ranges from 10 m to 108 m (from uncertainties in $H_{mc}$).

The bed topography data (Fig. 2b) indicates the essentially marine character of the Fleming Glacier, showing two basins featuring retrograde slopes, both located underneath the main trunk of the Fleming Glacier's fast flow region. The basin further upstream (hereafter "FG upstream basin") has a steeper retrograde slope than the one closer to the grounding line (hereafter "FG downstream basin"). For the FG downstream basin, elevation differences between bed_bm and the other two datasets (Figs 2c, 2d) show that bed_bm has a generally steeper retrograde slope. The sensitivity of basal friction coefficient distributions to the three bed datasets is discussed in Sect. 4.2.

### 2.3 Surface velocity data in 2008

The surface velocity data used for 2008 (Fig. 1b) were obtained from MEaSUREs InSAR-based Antarctic ice velocity (from the fall 2007 and/or 2008) produced by Rignot et al. (2011b) (version 1.0) with a resolution of 900 m and with uncertainties ranging from 4 m yr$^{-1}$ to 8 m yr$^{-1}$ over the study area. For the regions without data (grey area in Fig. 1b), we prescribe the surface speed to be 0. We do not use the finer (450 m) resolution MEaSUREs velocity here since the coarser (900 m) resolution data have been subjected to some post-processing, including smoothing and error corrections.

## 3 Method

All the simulations are carried out using the Elmer/Ice model (Gagliardini et al., 2013). These simulations solve the ice momentum balance equations with an inverse method to determine the basal friction coefficients, and the steady state heat equation to model the ice temperature distribution. The ice rheology is given by Glen's flow relation (Glen, 1955):

$$\boldsymbol{\tau} = 2\eta\dot{\boldsymbol{\varepsilon}} \tag{3}$$

where $\boldsymbol{\tau}$ is the deviatoric stress and $\dot{\boldsymbol{\varepsilon}}$ is the strain rate tensor. The viscosity $\eta$ is computed as:

$$\eta = \frac{1}{2}(EA)^{-1/n}\dot{\varepsilon}_e^{(1-n)/n} \tag{4}$$

where $E$ is an overall flow enhancement factor, A is a temperature-dependent rate factor calculated using an Arrhenius equation (Gagliardini et al., 2013), $\dot{\varepsilon}_e = \sqrt{tr(\dot{\varepsilon}^2)/2}$ is the effective strain rate, and $n$ is the exponent in Glen's flow law. Table 1 lists the parameters used in this study.

### 3.1 Mesh generation and refinement

We use GMSH (Geuzaine and Remacle, 2009) to generate an initial 2-D horizontal footprint mesh with the boundary defined from the grounding line data in 1996 (Rignot et al., 2011a) and the catchment boundary of the feeding glacier system (Cook et al., 2014), with the assumption that the ice front position in 2008 coincided with the grounding line position in 1996 (Rignot et al., 2011a). This assumption is tested as part of the sensitivity tests to various ice front positions.

To reduce the computational cost without reducing the accuracy, we refine the mesh with the anisotropic mesh adaptation software YAMS (Frey and Alauzet, 2005) using the local Hessian matrix (second order derivatives) of the surface velocity data in 2008 from Rignot et al. (2011c) as a metric for the mesh density. The resulting mesh is shown in Fig. 3 and has minimum and maximum element sizes of approximately 250 m and 4 km, respectively. The 2-D mesh is then vertically extruded using 10 equally spaced, terrain following layers. Sensitivity tests have been done on the Vestfonna ice cap (Schäfer et al., 2014; Schäfer et al., 2012) to demonstrate the robustness of inverse simulations to the vertical mesh resolution. In the current study an experiment with 20 extruded layers (not shown) gives very similar results as with 10 layers, confirming those findings also apply to the WIS-FG system. Experiments with various horizontal resolutions (1 km, 500 m, 250 m, and 125 m) show that 250 m are sufficient for the simulations on the WIS-FG system.

### 3.2 Boundary Conditions

For transient simulations (surface relaxation, section 3.3), the stress-free upper surface is allowed to evolve freely, with a minimum imposed ice thickness of 10 m over otherwise ice-free terrain. For inverse and temperature simulations, the upper surface height and temperature are fixed.

The surface temperature is defined by the yearly averaged surface temperature over 1979-2014 computed from the regional atmospheric climate model RACMO2.3/ANT27 (van Wessem et al., 2014). The geothermal heat flux (GHF) at the bed is obtained from Fox Maule et al. (2005) using input data from the SeaRISE project, and the GHF is interpolated with bilinear interpolation method from the standard 5 km grid onto the mesh. A basal heat flux boundary condition combining GHF and basal friction heating is imposed for temperature simulations.

At the ice front, the normal component of the stress where the ice is below sea level is equal to the hydrostatic water pressure exerted by the ocean. We will discuss the sensitivity to the ice front boundary condition in Sect. 4.4. On the lateral boundary, which falls within glaciated regions, the normal component of the stress vector is set equal to the ice pressure exerted by the neighboring glacier ice while the tangential velocity is assumed to be zero.

The bedrock is regarded as rigid, impenetrable, and temporally fixed in all simulations. The present-day solid Earth deformation rate in the Fleming glacier region (Zhao et al., 2017) is negligible compared to the uncertainty of the bedrock data. Assuming that basal melt is negligible under grounded ice, the normal basal velocity is set zero at the ice/bed interface. The sliding relation relates the basal sliding velocity $u_b$ to basal shear stress $\tau_b$. Considering that in this diagnostic study the sliding law is only used as a numerically convenient tool for calculating the basal shear stress, a simple linear sliding law following Gagliardini et al. (2013); Gillet-Chaulet et al. (2012) is applied on the bottom surface:

$$\tau_b = C u_b \tag{5}$$

where $C$, the basal friction coefficient, is used as the adjustable parameter in the inversion scheme described below. During the initial surface relaxation, and at the start of the inversion, $C$ is initialized to a constant value of $10^{-4}$ MPa m$^{-1}$ yr (following Gillet-Chaulet et al. (2012)), which is replaced with the inverted $C$ in subsequent steps.

### 3.3 Surface relaxation

There may be non-physical spikes in the initial surface geometry, caused for example by observational uncertainties of the surface or bedrock data and/or by the resolution discrepancy between mesh and geometry data. To reduce these features, we relax the free surface of this domain during a short transient simulation of 0.2 yr length with a timestep of 0.01 yr. This is long enough to remove the non-physical spikes, but too short to significantly modify the geometry of the fast flowing regions of the Fleming Glacier.

### 3.4 Inversion for basal friction coefficient

After the surface relaxation, we use a variational inverse method (MacAyeal, 1993; Morlighem et al., 2010) implemented in Elmer/Ice (Gagliardini et al., 2013; Gillet-Chaulet et al., 2012) to constrain the basal friction coefficient $C$ in Eq. (5). To avoid non-physical negative values, we use a logarithmic representation of the basal friction coefficient, $C = 10^{\beta}$, where $\beta$ can take any real value.

The inverse method is based on adjusting the spatial distribution of the basal friction coefficient to minimize a cost function that represents the mismatch between the magnitudes of the simulated and observed surface velocities:

$$J_0 = \int_{\Gamma_s} \frac{1}{2} (|\boldsymbol{u}| - |\boldsymbol{u}^{obs}|)^2 d\Gamma \tag{6}$$

where $\Gamma_s$ is the upper surface of the domain, $\boldsymbol{u}$ and $\boldsymbol{u}^{obs}$ are the simulated and observed surface velocities, respectively. We do not try to fit velocity directions.

To avoid over-fitting of the inversion solution to non-physical noise in the observations, a regularization term is added to the cost function as:

$$J_{tot} = J_0 + \lambda J_{reg} \qquad\qquad\qquad (7)$$

where $J_{reg}$ is a regularization term imposing a cost on spatial variations in the control parameter $\beta$, $\lambda$ is a positive regularization weighting parameter, and $J_{tot}$ is the total cost (following for example Gillet-Chaulet et al. (2012)). Thus, the minimum of this cost function is no longer the best fit to observation but a compromise between fit to observation and smoothness in $\beta$. An L-curve analysis (Hansen, 2001) has been carried out for inversions in the current study to find the optimal $\lambda$ by plotting the term $J_{reg}$ as the function of $J_0$ (Fig. S2 in the supplementary material). The optimal value of $10^8$ is chosen for $\lambda$ to minimize $J_0$.

**3.5 Steady-state temperature simulations**

In the absence of a known englacial temperature distribution for the Fleming Glacier system, the steady state heat transfer equation is solved using an iterative method as described in Gagliardini et al. (2013) to provide temperatures for use in the inversion process. The ice velocity and geometry are held constant for this part of the simulation. Steady-state temperature simulations for a non-steady-state glacier system will result in estimations of temperatures that deviate from reality. Similar experiments on the Greenland Ice Sheet indicated that the simulated steady-state temperature field could provide a reasonable thermal regime for calculation of basal conditions (Seroussi et al., 2013).

**3.6 Experiment design**

Gong et al (2017) adopted a four-step spin-up scheme (Gladstone et al., 2014) in inverse modelling using Elmer/Ice (Gagliardini et al., 2013), without testing the effect of assumptions about the initial englacial temperature distribution on the inversion results. To explore the sensitivity of inverse modeling to initial temperature assumptions, we proposed a spin-up scheme with more cycles (three cycles in this study as presented in Fig. 4). For each cycle, we followed the spin-up scheme from Gladstone et al. (2014):

1. surface relaxation;
2. inversion of the basal friction coefficient using the relaxed surface geometry;
3. a steady state temperature simulation using the simulated velocities from that inversion;
4. another inversion with the previously obtained steady-state temperature.

The surface relaxation for each cycle starts from the same initial geometry described in Sect. 3.3. For cycle 1, the surface relaxation and first inversion are implemented with an initial temperature assumption (described below) and uniform basal friction coefficient of $10^{-4}$ MPa m$^{-1}$ a (following Gillet-Chaulet et al. (2012)). For cycles 2 and 3, the surface relaxation and inversion are initiated with the simulated steady-state temperature and an initial distribution of basal friction coefficient $C$ from the final state of the previous cycle.

To explore the sensitivity of our inverse method to assumed initial englacial temperature distribution, enhancement factor ($E$), basal topography, ice front positions, and the ice front boundary conditions, we carry out the experiments summarized in Table 2.

An assumed initial englacial temperature distribution is used in the first cycle of the scheme above and would affect the surface relaxation, the modelled ice deformation and the ice velocity field, especially for fast-flowing regions, and consequently affect the steady-state temperature calculation, which might affect the subsequent inversion process. To explore the impact of initial temperatures on inversion results with the three-cycle spin-up scheme, we propose experiments with different initial temperature assumptions for the surface relaxation and initial inversion in Cycle 1. TEMP1: a uniform temperature of -20 ℃; TEMP2: a uniform temperature of -5 ℃; CONTROL: a linearly increasing temperature from the upper surface values (see also Sect. 3.2) to the pressure melting temperature at the bed. To test the sensitivity of basal friction to the relaxed geometry, we also add another experiment -

TEMP3: surface relaxation in the first cycle using the linear temperature, followed by inversion with a uniform temperature of -20 ℃. Experiments TEMP1, TEMP2 and TEMP3 differ from CONTROL only in the temperature fields imposed before the first temperature simulation.

Ma et al. (2010) tested the influence of ice anisotropy on the ice flow through various enhancement factors, and found that appropriate $E$-values for the grounded ice are usually >1.0. To find out the most appropriate value of $E$ (in Eq. 4) in this study, we evaluate inversion carried out with different values of $E$ (EF1: $E = 0.5$, CONTROL: $E = 1.0$, EF2: $E = 2.0$, EF3: $E = 4.0$; Table 2). Experiments EF1, EF2 and EF3 differ from CONTROL only in terms of the value used for $E$.

As described in Sec. 2.2, we generate three different bed topography datasets to explore the sensitivity of the inverse modelling. The three-cycle spin-up scheme is carried out for each bed dataset using the linear initial temperature distribution described above. These experiments are referred to as CONTROL, BEDZC, and BEDMC (Table 2). Experiments BEDZC and BEDMC differ from CONTROL only in terms of the bedrock data set used.

In our standard model domain we assume the 2008 ice front is coincident with the 1996 grounding line, which has an error of several km on fast-moving ice (Rignot et al., 2011a) and might have changed since 1996. The frontal surface elevation is from the SPOT DEM data in Jan 2008, which shows the ice front position is ~1.5 km downstream of the 1996 grounding line position. Since such a narrow residual ice shelf is considered unlikely to have a major
influence, we construct the model geometry to have the ice front coincide with the 1996 grounding line for simplicity, i.e. all ice is considered grounded. To test the sensitivity of inverse modelling to the ice front positions, we implement two further scenarios with different ice front positions: downstream (experiment IFP1) and upstream (experiment IFP2) of the 1996 grounding line position (CONTROL). In IFP1, we assume the ice front position is
coincident with the frontal boundary of SPOT DEM data (~1.5 km downstream). In IFP2, we artificially put the ice front position ~1.5 km upstream of the 1996 grounding line position for ~1.5 km. IFP1 and IFP2 differ from CONTROL only in their ice front position.

    In addition to the ice front position, there are other sources of uncertainty in the vicinity of the ice front: ice thickness, bedrock depth, height conversion from geoid to ellipsoid, and
backstress due to the presence of ice mélange. These uncertainties have an effect on the pressure boundary condition applied to the ice front, which conventionally balances the normal stress in the ice against ocean water pressure. In view of the ice thickness uncertainty (ranging from 10 m to 100 m) and hence bedrock depth around the grounding line, and given the possibility of increased additional buttressing force due to floating icebergs and ice
mélange as indicated in many previous studies (e.g. Amundson et al. (2010); Krug et al. (2015); Robel (2017); Todd and Christoffersen (2014); Walter et al. (2017)) and clearly seen in Fig. 1c, we vary the ocean pressure boundary condition by varying the sea level used to calculate ocean water pressure. This approach directly represents some small uncertainty in the actual sea level, but is also a proxy for pressure variations due to bedrock elevation/ice
thickness uncertainty and mélange back stress. First in the CONTROL experiment, we assume an ocean pressure at the ice front computed using the observed sea level of 15 m, as mentioned in Sec. 2.1. We further simulate two alternative scenarios for the sea level used in the simulations to calculate ocean pressure: IFBC1 with a sea level of 5 m and IFBC2 with a sea level of 25 m. Another extreme scenario (IFBC3, Table 2) is adopted here by setting the
ice front pressure to the ice overburden:

$$P_i(z) = \rho_i g(z_s - z) \qquad\qquad\qquad\qquad\qquad (7)$$

where $P_i(z)$ is the pressure at the ice front as a function of height $z$, $\rho_i$ is ice density (Table 1), $g$ is the gravitational constant (Table 1), and $z_s$ is the height of ice upper surface at the ice front. This is the pressure that would be imposed by a hypothetical undeforming continuation
of the advancing glacier, and imposes zero normal strain rate at the ice front. The ice surface elevation $z_s$ at the front is ~115 m, approximately 100 m above actual sea level. The total

vertically integrated pressure imposed by this condition is equivalent to a sea level of ~60 m, although the vertical distribution of pressure differs from an ocean pressure condition. Experiments IFBC1, IFBC2 and IFBC3 differ from CONTROL only in their ice front boundary condition.

## 4 Results and discussions

The main focus of the current study is the sensitivity of the inversion to the variations of five factors: temperature initialization, enhancement factor, bed topography, ice front positions, and ice front oceanic pressure boundary condition. The evaluation criteria are the robustness of simulated basal friction coefficient distribution to experiment design and the mismatch between the simulated and observed surface velocities.

### 4.1 Sensitivity to initial temperature

We present the results for the inferred basal friction coefficients from the CONTROL and three TEMP experiments (Sect. 3.6, Table 2) for the WIS-FG system in Fig. 5. The 2008 ice velocity contours are added as visual references for comparing the basal friction coefficient patterns in the regions of fast flow, since the largest observed ice velocity changes occurred in fast flowing outlet regions (Mouginot et al., 2014; Walker and Gardner, 2017).

In each cycle, the root-mean-square deviation (RMSD, sometimes also called root-mean-square error) between the relaxed and the observed surface was < 25 m (see Table S1 in the supplementary material), smaller than the ice thickness uncertainty (> 50 m) used in this study. However, the systematic changes generated at the ice front during the surface relaxation may have an effect on the inversion, and this is further discussed in Sect. 4.4.

After the first cycle (left column, Fig. 5), results showed different patterns of basal friction coefficient for each experiment, especially in the fast-flowing regions with surface velocity exceeding 1000 m yr$^{-1}$ (yellow contour in Fig. 5). The basal friction coefficients from TEMP2 (Fig. 5g) and CONTROL (Fig. 5a) share similar sticky spots around the ice front, and some isolated sticky spots ~3-5 km upstream of the ice front, but TEMP1 (Fig. 5d) and TEMP3 (Fig. 5j) display different patterns, indicating dependence on the initial temperature assumption. The RMSDs of key properties are computed to evaluate the consistency of these experiments (Tables 2, S2-S5).

To reduce the dependence on initial temperature and achieve a consistent equilibrium thermal regime with respect to the given friction coefficient distribution, we carried out the second cycle shown in Fig. 4. The basal friction coefficients from the final step of Cycle 2 (the middle column in Fig. 5) show greater similarity across all the temperature experiments. However, for experiments CONTROL and TEMP2, the isolated sticky points ~3-5 km upstream of the ice front (with horizontal scale around ~1 km and peak basal friction coefficient of around $6\times10^{-5}$ MPa m$^{-1}$ yr) mostly decrease or disappear from the first cycle (Figs. 5a, 5g) to the second cycle (Figs. 5b, 5h). Therefore, a third cycle was implemented to test whether a two-cycle spin-up scheme was enough to reduce the dependence on the initial temperature assumptions. After the third cycle, all the scenarios depicted a similar basal friction coefficient pattern (right column in Fig. 5). These differences in basal friction coefficients between the TEMP simulations can also be analyzed through Table S2 and Fig. S4. While these statistics and visualizations confirm the similarity between CONTROL, TEMP2 and TEMP3, it is evident that TEMP1 still shows notable differences to these simulations, even after three cycles (see also Table S3 for basal velocity RMSD). The CONTROL simulation, starting with a linear interpolation of temperature from upper to lower surfaces, seems to be the best option for several reasons: the choice of temperature value for upper and lower surfaces is physically motivated, which is not true for the other assumptions; it shows the lowest RMSD between simulated and observed upper surface velocity of the

temperature sensitivity simulations (Table 2); and it shows the least change in the temperature
distribution over the three cycles (Table S4). Given this choice of preferred temperature
initialization (CONTROL), and the significant difference between this and the cold
initialization (TEMP1), we argue that TEMP1 likely deviates furthest from an ideal
temperature initialization, and that such a large initial deviation would require more than three
cycles to converge on a basal friction coefficient distribution. In other words, we postulate
that the three cycles are likely sufficient to provide a robust inversion only for initial
temperatures moderately close to reality, with the linear interpolation in the vertical providing
the most appropriate initial guess amongst our tests. Hence, we adopted the scenario with
initial linear temperature for the experiments described hereafter.

The present study is focused on exploring the effects of uncertainties and their control, while
the dynamics of the FG system will be discussed in more detail in a companion paper (Zhao
et al., companion paper). However, a few comments are in order regarding the contrast with
an earlier study on the Vestfonna ice cap. The low impact of temperature profile on the basal
friction coefficient distribution in that study was due to a lower contribution of ice
deformational motion compared to basal sliding (Schäfer et al., 2012). Internal ice
deformation, and hence temperature, may be especially important for the WIS-FG system due
to steep surface slopes and corresponding high driving stresses in the region between the
downstream and upstream basins (~8-12 km upstream of the ice front in Fig. S5a). The
patterns of basal friction coefficient (right column of Fig. 5) all indicate substantial spatial
variation in basal friction over the fast flowing part of the FG. For example, in the region
flowing faster than 1000 m yr$^{-1}$ (inside the yellow contour), we see very low friction over the
downstream basin, but higher friction coefficients over the upstream bedrock high, and in a
narrow band along the ice front. A comparison between the simulated basal and surface
velocities (Fig. S5b) shows that vertical shear dominates the ice dynamics in the region of
high slope between the downstream and upstream basins, where the driving stress is relatively
high. This alone would suggest a high sensitivity of modelled sliding velocity and basal
friction to the englacial temperature.

The multi-cycle iterative spin-up scheme is suggested as an effective set-up for inverse
modelling of fast-flowing glaciers that have high surface slopes and vertical shear strain rates
and therefore are sensitive to the internal vertical ice temperature distribution. In the present
application to the Fleming system, three cycles were sufficient, except in the case of an
unphysically cold initialization. In other cases, the inversion process is not so heavily
dependent on the temperature field, for example for reproducing the shear margins of the
outlet glacier of Basin 3 on Austfonna ice cap, Svalbard (Gladstone et al., 2014).

**4.2 Sensitivity to enhancement factor**

Sensitivity of inverse modelling to the flow enhancement factor has been explored by
experiments EF1-3 and the results (after three-cycle procedure) are shown in Fig. 6. The
simulated basal friction coefficients (left column in Fig. 6) show different patterns with
different $E$ values. Recall that from Eq. (4), smaller $E$ means higher ice viscosity. The local
high friction coefficient sticky spots near the ice front expanded both upstream and along the
ice front with increased $E$ values, forming a band across the ice front for $E = 4.0$ (EF3).
Conversely, inversions with smaller $E$ give a better-simulated surface velocity at the ice front
(middle column in Fig. 6), and also lead to smaller differences between the observed and
relaxed surface elevation after the surface relaxation (right column in Fig. 6), whereas for EF3
the surface relaxation generates a considerable steepening of the surface slope towards the ice
front (Fig. 6l). However, the computed RMSD of the surface velocity mismatch for the fast
flowing regions (> 1500 m yr$^{-1}$, middle column in Fig. 6 and Table 2) indicates that the
experiment EF1 ($E = 0.5$) (Fig. 6e) shows greater underestimation of surface velocity than
CONTROL (Fig. 6f). Therefore, the optimal value of $E = 1.0$ is chosen as the most suitable
enhancement factor for the Fleming system.

## 4.3 Sensitivity to bedrock topography

Figure S6 summarizes results from the three experiments using different bed topographies (Sect. 3.7, Table 2). As shown in Fig. S6, the simulated basal friction coefficient $C$ varies slightly with bedrock geometry and its distribution shows greater similarity between BEDZC and BEDMC, compared with CONTROL. CONTROL (using Bedmap2 bedrock data; Fig. S6a) shows slightly smaller basal friction coefficients than BEDMC (Fig. S6b) and BEDZC (Fig. S6c) in the fast-flowing region ($>1500$ m yr$^{-1}$, cyan contour in Fig. S6). The pattern in the region between the 1000 and 1500 m yr$^{-1}$ contours also differs compared to the CONTROL case, which might be caused by the deeper bedrock of Bedmap2 in this region (Fig. S6g), compared to the other two datasets (Figs. S6h, S6i). However, all three cases feature a low basal friction coefficient in fast flow regions ($>1500$ m yr$^{-1}$ in Fig. S6), which is approximately coincident with the FG downstream basin.

The simulated surface velocities from BEDZC (Fig. S6e) and BEDMC (Fig. S6f) match the observed surface velocities better than those from CONTROL (Fig. S6d) in the regions around the ice front and more broadly for velocity exceeding 1000 m yr$^{-1}$. This point is supported by the computed RMSD of surface velocity mismatches (Table 2). One possible cause of the different basal friction coefficient distributions in these inversions might be the changed surface topography during the surface relaxation, especially near the ice front (Figs. S7).

Comparisons of the distributions of velocity mismatch and of $C$ between BEDZC and BEDMC do not provide a direct insight into which is the more accurate basal geometry for modelling the Fleming system. The computed RMSD of the velocity mismatch for the regions with velocity $>1500$ m yr$^{-1}$ (Table 2) is only slightly higher for BEDMC (62.60 m yr$^{-1}$) than for BEDZC (61.78 m yr$^{-1}$), and both are much lower than CONTROL. Both BEDMC and BEDZC use the 2008 surface DEM and this improvement over the Bedmap2 surface DEM in CONTROL appears significant, even before turning to the matter of ice thickness. Both cases use the ice thickness extracted using the mass conservation approach (which is independent of surface geometry) and the bed geometries are accordingly more similar to each other than they are to CONTROL (see Fig. 2b-d). However, BEDZC maintains better internal consistency with the 2008 surface elevation, since it results in the mass conserving ice thickness $H_{mc}$ being employed, whereas, by the construction of bed_mc (Eq. 2), the ice thickness in BEDMC is not entirely consistent with mass conservation, although still a more physically motivated interpolation than bed_bm in CONTROL. The BEDMC and BEDZC ice thicknesses clearly differ by the difference between the Bedmap2 and 2008 DEMs, which should be greatest in areas of greatest lowering, and as we see BEDMC provides a useful sensitivity test case. Since bed_zc is extracted from the accurate and contemporary DEM2008, it should also incorporate into the bed geometry (via $H_{mc}$) more detail from the then current surface, compared to bed_mc, extracted from Bedmap2's surface DEM, which was generated over a longer time range. Therefore, bed_zc is suggested as the best current bedrock elevation data for further ice sheet modelling of the WIS-FG system.

## 4.4 Sensitivity to ice front position and boundary condition

All the inversions presented so far feature some sticky spots with high basal friction coefficient near the ice front of the Fleming Glacier (right column of Fig. 5 and left column of Fig. S6). We now consider causes for possible uncertainties in the force applied to the ice front, and whether high basal friction near the ice front is likely to be a feature of the real system or emerges from the inversion process as a compensating response to incorrect boundary forcing. These possible causes include uncertainty in local bedrock elevation (or equivalently ice thickness), uncertainty in the geoid-ellipsoid height conversion, uncertainty in observed sea level, uncertainty in exact ice front position and grounding line position, uncertainty in surface velocity, and uncertainty in potential backstress due to ice mélange and/or grounded icebergs in contact with the ice front. The sensitivity to various bedrock

datasets has been discussed in Sec. 4.3. By assuming the ice front position to coincide with the 1996 grounding line, uncertainty about the bedrock depth at the ice front feeds into significant uncertainty in the total restraining force from ocean pressure. Regarding velocities,

Friedl et al. (2018) presented evidence that an acceleration phase occurred on the Fleming Glacier between Jan-Apr 2008, but the surface velocity data used in the current study was extracted from measurements in Fall 2007 and 2008 (Rignot et al., 2011b). This means the surface velocity data, which provide the target to be matched by the inversion, might not be consistent with the DEM data used here (acquired in Jan 2008). To explore the influence of

these different sources of uncertainty, we adopt different ice front positions and effective sea level heights as described in Sect. 3.6 (IFBC1-3 and IFP1-2, Table 2).

Experiments with different ice front positions (IFP1-2 in Table 2) directly affect the ice thickness and bed elevation at the ice front, which affects the ice front pressure condition. The simulated basal friction coefficients (left column in Fig. 7) show that the high sticky spots

near the ice front migrate with the ice front position but with different patterns. The experiment IFP1 with a seaward shifted ice front position shows a decrease in magnitude of the high friction spots (Fig. 7b) and a better match with the observed velocity (Fig. 7e), while the IFP2 with a retreated ice front shows an increased $C$ (Fig. 7c) and worse surface velocity match (Fig. 7f) compared with CONTROL experiment (Figs. 7a, 7d). In experiment IFP1,

thinner ice at the ice front leads to a relatively smaller ice velocity compared with CONTROL, so the model does not need to increase $C$ to match the observed surface velocity. This does not mean that ice front position in IFP1 is more accurate than CONTROL, since the time inconsistency of surface DEM data, ice front and grounding line position, and surface velocity data is the obstacle to obtaining a reliable basal friction pattern. Therefore, we speculate that

some of the high basal friction spots near the ice front are artefacts. However, we do not exclude the possibility of high basal friction spots caused by the pinning points located at the 1996 grounding line, which is also proposed by Friedl et al. (2018). An accurate location of the ice front and grounding line is clearly important for inverse modelling of fast flowing glaciers like the Fleming Glacier.

A higher sea level in the ice front boundary condition imposes a higher pressure at the ice front, i.e. a higher total retarding force, and we impose these different boundary conditions as a proxy for the sources of uncertainty discussed above. Basal friction coefficients $C$ simulated from the IFBC1-2 and CONTROL experiments (Figs. 8a-c) present similar patterns but differ systematically around the ice front regions (within ~1 km of the grounding line). Experiments

with higher sea levels display smaller $C$ there (Fig. 8, left column) and provide a better match between modeled and simulated surface velocities (Fig. 8, right column), which is consistent with the computed RMSD of the surface velocity mismatch (Table 2). If the applied ice front boundary condition underestimates the real world forcing, the inversion process will compensate by increasing the basal friction in this region.

Experiment IFBC3, with an extreme assumption of applying ice pressure corresponding to a neighboring column of ice matching the ice front, shows very small basal friction for the downstream basin (Fig. 8d). However, IFBC3 introduces a much greater mismatch to the observed surface velocities, with underestimated velocities over a substantial region extending upstream from the ice front and greater overestimate of velocities further upstream.

This is only a sensitivity test but implies a potentially suitable ice front pressure may lie between IFBC2 and IFBC3. This set of experiments also suggests that moderate changes influence only a limited area. It is hard to decide the best ice front boundary condition here owing to the lack of precise bedrock data (as seen above) and difficulty of estimating the additional pressure from the partly detaching icebergs and ice mélange. As an indicator, the

simulated ice mélange depth-integrated back stress (~$1.1\times10^7$ N m$^{-1}$) required to prevent the iceberg rotation at a calving front (Krug et al., 2015) would be comparable to an additional ~2.3 m in sea level in terms of ice front boundary condition for the Fleming Glacier. The thickness and density of mélange may affect this estimation. But it is certainly clear that the

ice front boundary conditions can have a significant effect on the inversion results near the grounding line.

The different ice front boundary conditions also lead to minor differences in the surface relaxation at the ice front, with lower sea levels leading to slightly greater lowering and corresponding steepening of the surface adjacent to the ice front (for example, ~8 m lowering from IFBC2 to CONTROL and from CONTROL to IFBC1 at the ice front). The differences in surface elevation are localized to the ice front zone, with the relaxation over the rest of the domain essentially unaffected, except for the most extreme forcing. The lowered surface at the ice front in experiments IFBC1 and CONTROL is apparently the consequence of rapid deformation due to its own weight (longitudinal extension with locally high vertical shear) of an ice cliff, which is over 100 m higher than the control sea level. However, the sticky spot located ~1 km upstream the ice front is a persistent feature except for the experiment IFBC3. This implies that the high friction near the ice front is sensitive to the boundary condition at the ice front.

Based on the experiments IFP 1-2 and IFBC1-3, we suspect the high friction near the ice front is likely an artefact due to errors in the ice front boundary condition but we cannot rule out the possibility that this may be a real feature. However, the impact diminishes rapidly with distance inland for moderate sea level shifts, which do not affect the general pattern of basal friction coefficients or the quality of the velocity matching more than ~2 km upstream of the grounding line.

**5 Conclusions**

We have obtained a basal friction coefficient distribution for the Wordie Ice Shelf-Fleming Glacier system in 2008, using an iterative spin-up scheme of simulations, observed surface velocities and a detailed surface DEM. We explored the sensitivity of the inversion for basal friction to four inputs to the modelling process. Within the approximation of using simulated steady-state ice temperatures, we showed that multiple temperature-inversion cycles are necessary to remove the influence of initial englacial temperature assumptions, at least for plausible initial temperature assumptions, and that a poor initial assumption will lead to a requirement for a greater number of cycles. This conclusion is expected to also apply to other fast-flowing glacier systems that feature high rates of internal deformation.

Our inversion of the Wordie Ice Shelf-Fleming Glacier system is highly sensitive to the choice of ice flow enhancement factors and basal elevation datasets. The "bed_zc" bed topography, which used ice thickness determined using the mass conservation method for the fast-flowing regions, using contemporary velocities and ice thinning rates, and applied to the then current DEM, is suggested as the best current bed topography for further simulations in this region.

For the Wordie Ice Shelf-Fleming Glacier system, which we treated as grounded adjacent to the ice front, the inferred basal friction coefficient near that ice front is sensitive to the ice front position and ocean pressure boundary condition, emphasizing the importance of the normal force on the ice front and the accuracy of ice front positions. These factors have a very low impact on basal friction coefficients more than a few kilometers upstream of the grounding line, but may still be important when using inversion to initialize transient simulations, due to the high sensitivity of transient ice dynamic behavior to grounding line dynamics.

**Author Contributions**

Chen Zhao and Rupert Gladstone designed the experiments together. Chen Zhao collected the datasets, ran the simulations, and drafted the paper. Mathieu Morlighem generated the mass-

conservation constrained ice thickness data. All authors contributed to the refinement of the experiments, the interpretation of the results and the final manuscript.

**Acknowledgements**

Chen Zhao is a recipient of an Australian Government Research Training Program Scholarship and Quantitative Antarctic Science Program Top-up Scholarship. Rupert Gladstone is funded by the European Union Seventh Framework Programme (FP7/2007-2013) under grant agreement number 299035 and by Academy of Finland grant number 286587. Matt A. King is a recipient of an Australian Research Council Future Fellowship (project number FT110100207) and is supported by the Australian Research Council Special Research Initiative for Antarctic Gateway Partnership (Project ID SR140300001). Thomas Zwinger's contribution has been covered by the Academy of Finland grant number 286587. This work was supported by the Australian Government's Business Cooperative Research Centres Programme through the Antarctic Climate and Ecosystems Cooperative Research Centre (ACE CRC). This research was undertaken with the assistance of resources and services from the National Computational Infrastructure (NCI), which is supported by the Australian Government. We thank Fabien Gillet-Chaulet for advice on implementation of the inversion. We thank E. Rignot, J. Mouginot, and B. Scheuchl for making their SAR velocities publicaly available. SPOT 5 images and DEMs were provided by the International Polar Year SPIRIT project (Korona et al., 2009), funded by the French Space Agency (CNES). The ASTER L1T data product was retrieved from https://lpdaac.usgs.gov/data_access/data_pool, maintained by the NASA EOSDIS Land Processes Distributed Active Archive Center (LP DAAC) at the USGS/Earth Resources Observation and Science (EROS) Center, Sioux Falls, South Dakota.

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

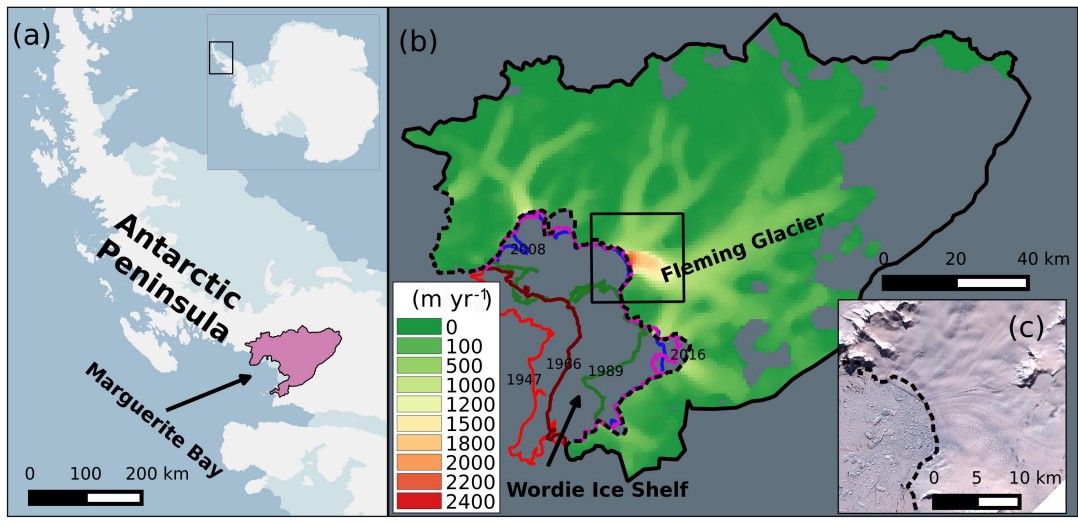

Figure 1. (a) The location of the Wordie Ice Shelf-Fleming Glacier system in the Antarctica Peninsula (pink polygon). (b) Surface speed in 2008 with a spatial resolution of 900 m obtained from InSAR data (Rignot et al., 2011c) for the study regions. Colored lines represent the ice front position in 1947 (red), 1966 (brown), 1989 (green), Apr 2008 (blue), and Jan 2016 (magenta) obtained from Cook and Vaughan (2010), Wendt et al. (2010), and Zhao et al. (2017). The grey area inside the catchment shows the region without velocity data. (c) Ice front images acquired from ASTER L1T data on Feb 2$^{nd}$, 2009. The dashed line in (b) and (c) is the 1996 grounding line position (Rignot et al., 2011a).

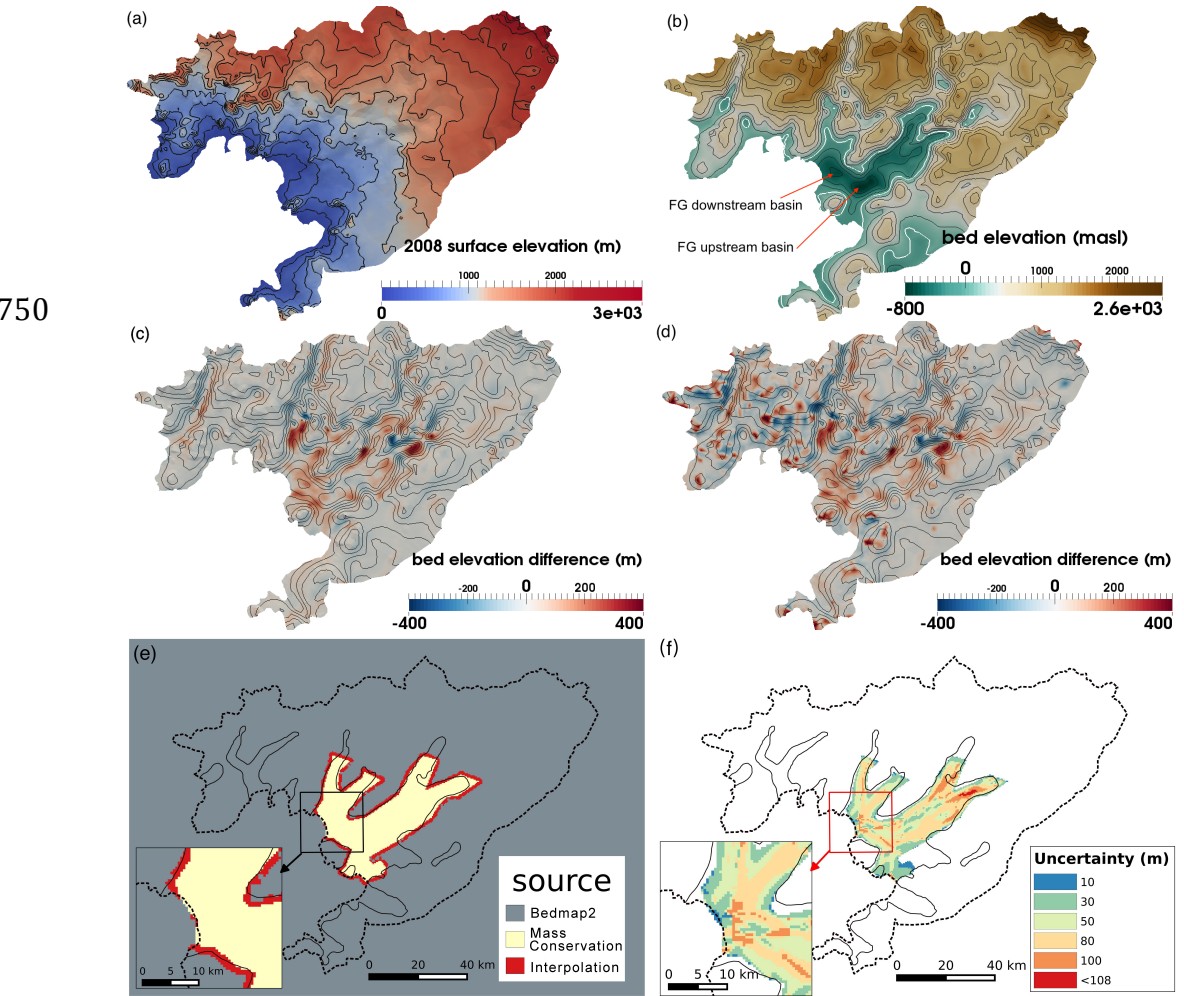

Figure 2. (a) Surface elevation data in 2008 with black contours (interval: 200 m) representing the surface elevation. (b) bed elevation data from bed_bm (meters above sea level, masl), (c) elevation difference between bed_mc and bed_bm (d) elevation difference between bed_zc and bed_bm. The black contours in (b-d) show the bed elevation with an interval of 200 m. (e) The ice thickness data sources and (f) the uncertainty of the ice thickness data $H_{mc}$ with black solid lines representing the observed ice surface velocity of 100 m yr$^{-1}$.



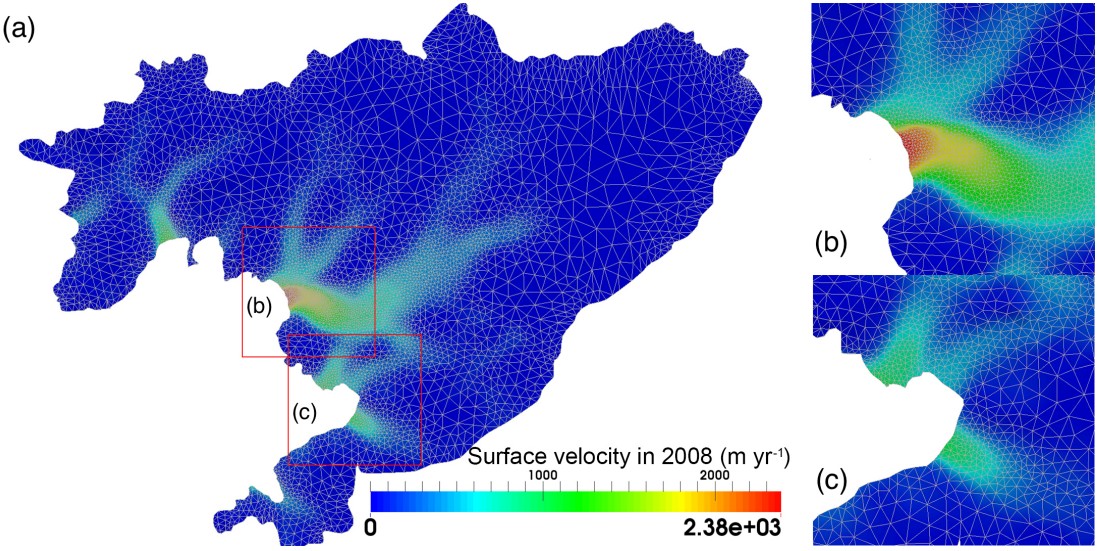

Figure 3. (a) Mesh structure of the domain in the current study with surface velocity in 2008 (Rignot et al., 2011c) and the zoomed-in map for (b) the Fleming Glacier and (c) the Prospect Glacier.

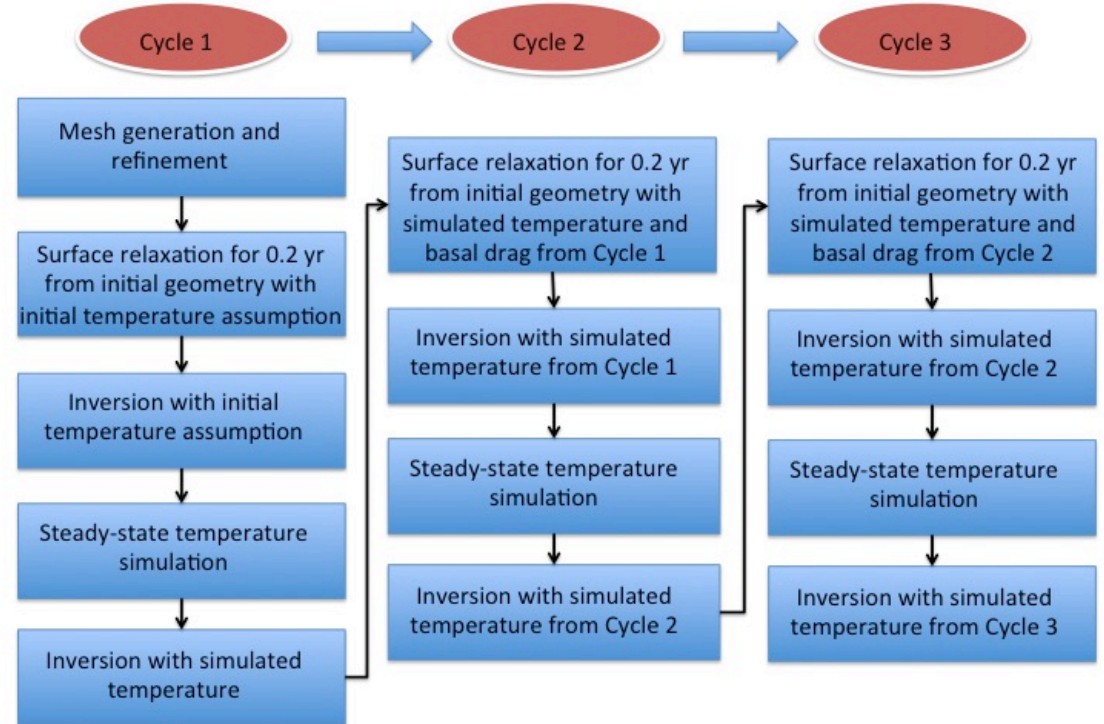

Figure 4. Flow chart of simulation spin-up with three cycles.

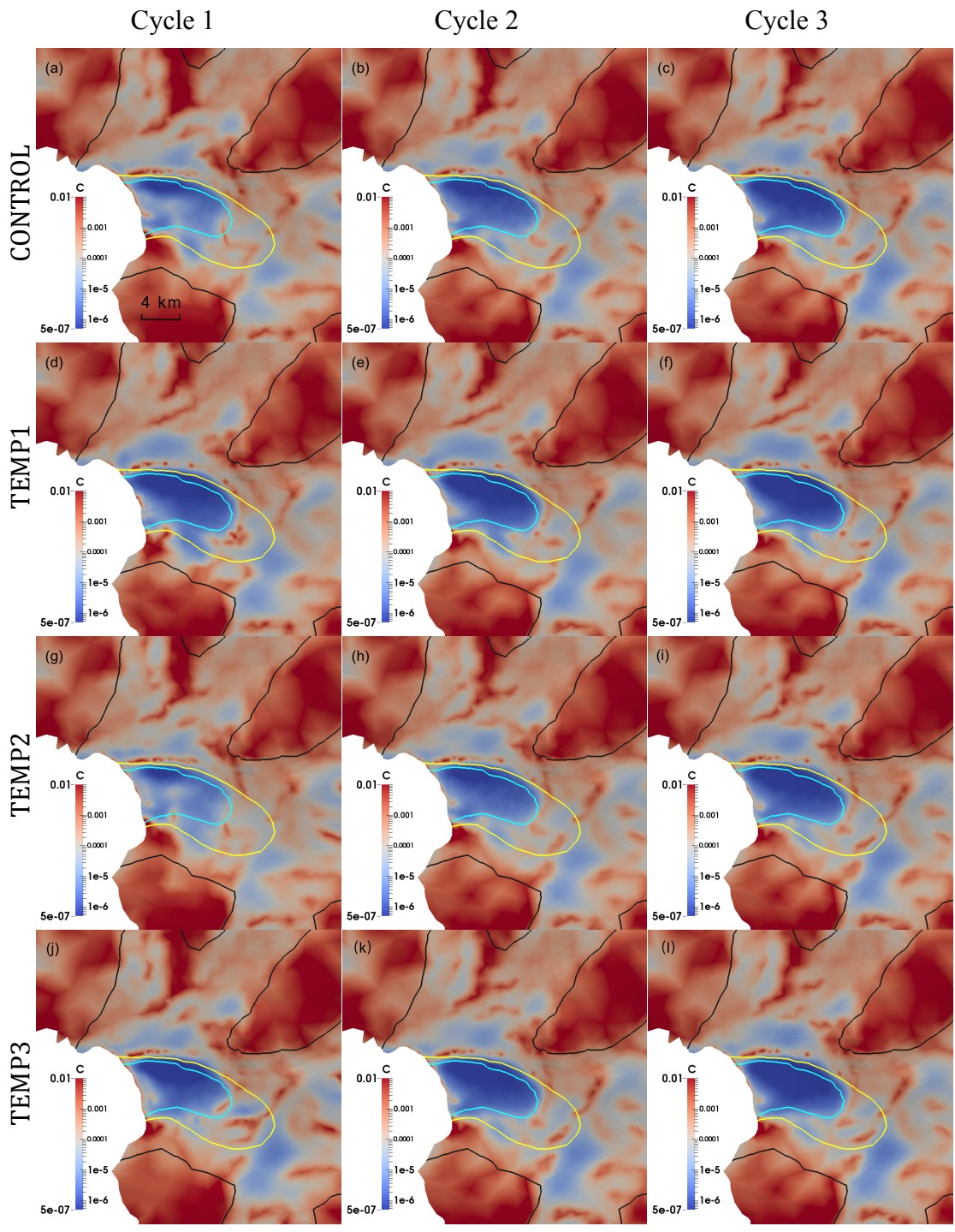


Figure 5. Basal friction coefficient $C$ (MPa m$^{-1}$ yr) inferred from experiments: (a-c) CONTROL (first row), and (d-f) TEMP1 (second row), (g-i) TEMP2 (third row), and (j-l) TEMP4 (forth row). The left (a, d, g, j), middle (b, e, h, k) and right columns (c, f, i, l) are the inferred basal friction coefficients from Cycle 1, Cycle 2 and Cycle 3, respectively. The

black, and yellow, and cyan solid lines represent observed surface speed contours of 100 m yr$^{-1}$ ,and 1000 m yr$^{-1}$ and 1500 m yr$^{-1}$, respectively.

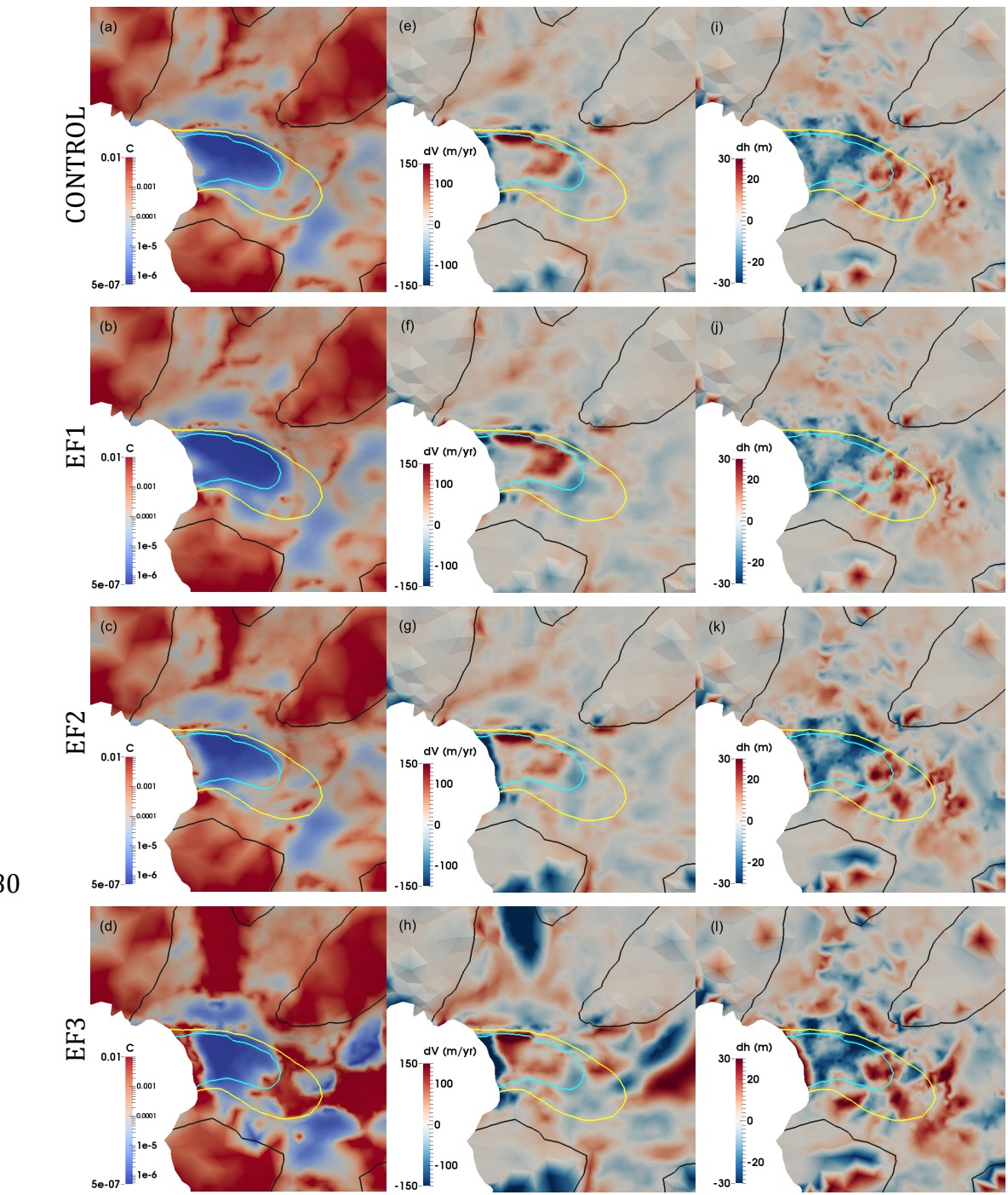

Figure 6. Distribution of basal friction coefficient $C$ (MPa m$^{-1}$ yr) (left column), mismatch between the observed and modeled surface velocity (observed minus simulated; middle column), and the difference between the observed initial surface and relaxed surface elevation (observed minus relaxed; right column) from experiments: (a, e, i) CONTROL (first row), (b, f, j) EF1 (second row), (c, g, k) EF2 (third row), and (d, h, l) EF3 (forth row). The black, yellow and cyan solid lines represent observed surface speed contours of 100 m yr$^{-1}$, 1000 m yr$^{-1}$, and 1500 m yr$^{-1}$, respectively.

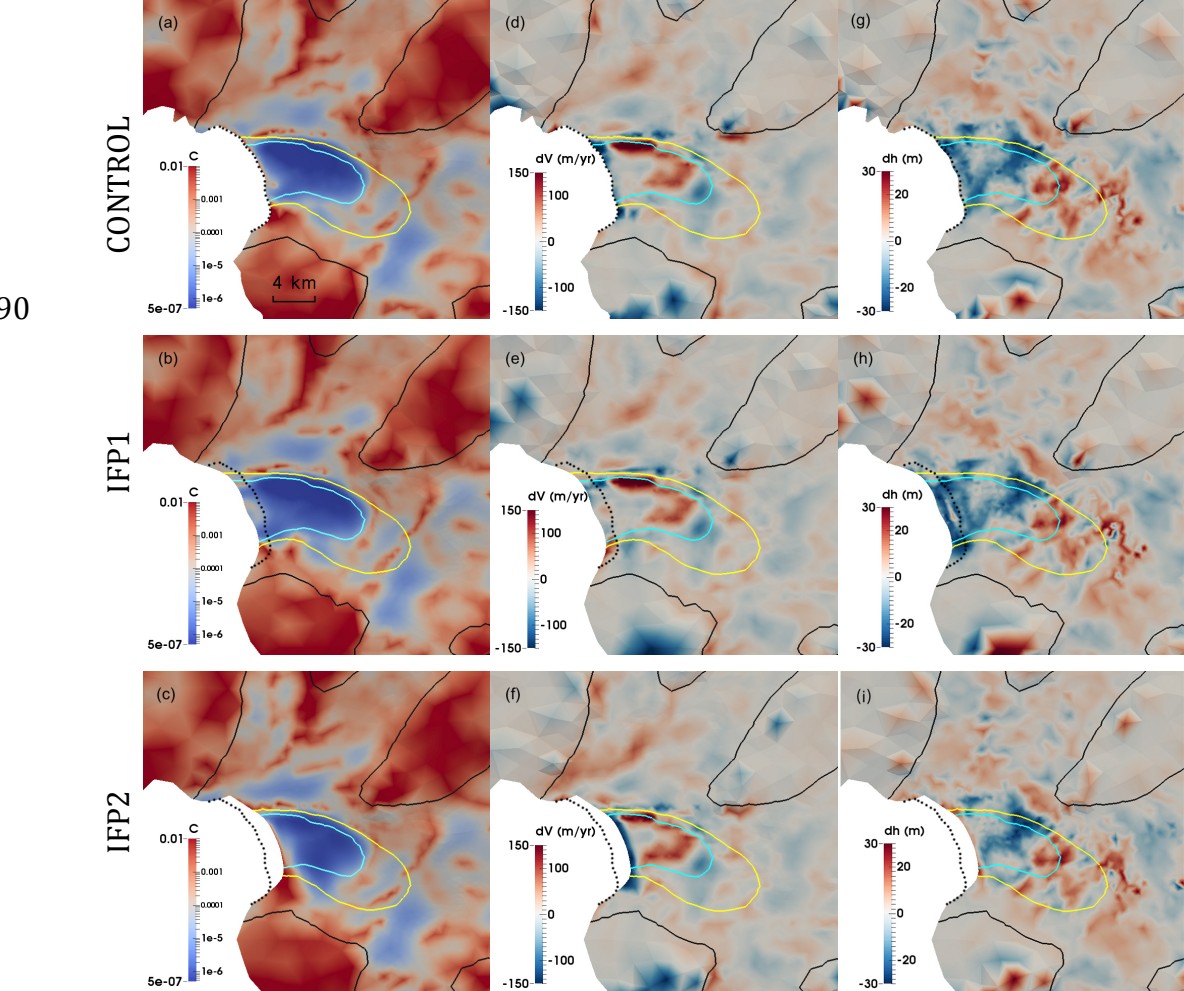

Figure 7. Distribution of basal friction coefficient C (MPa m$^{-1}$ yr) (left column), the mismatch between the observed and modeled surface velocity (observed minus simulated; middle column), and the difference between the observed initial surface and relaxed surface elevation (observed minus relaxed; right column) from experiments: (a, d, g) CONTROL (first row), (b, e, h) IFP1 (second row), and (c, f, i) IFP2 (third row). The black, yellow, and cyan solid lines represent surface speed contours of 100 m yr$^{-1}$, 1000 m yr$^{-1}$, and 1500 m yr$^{-1}$, respectively. Black dotted line is the 1996 grounding line position (Rignot et al., 2011a).

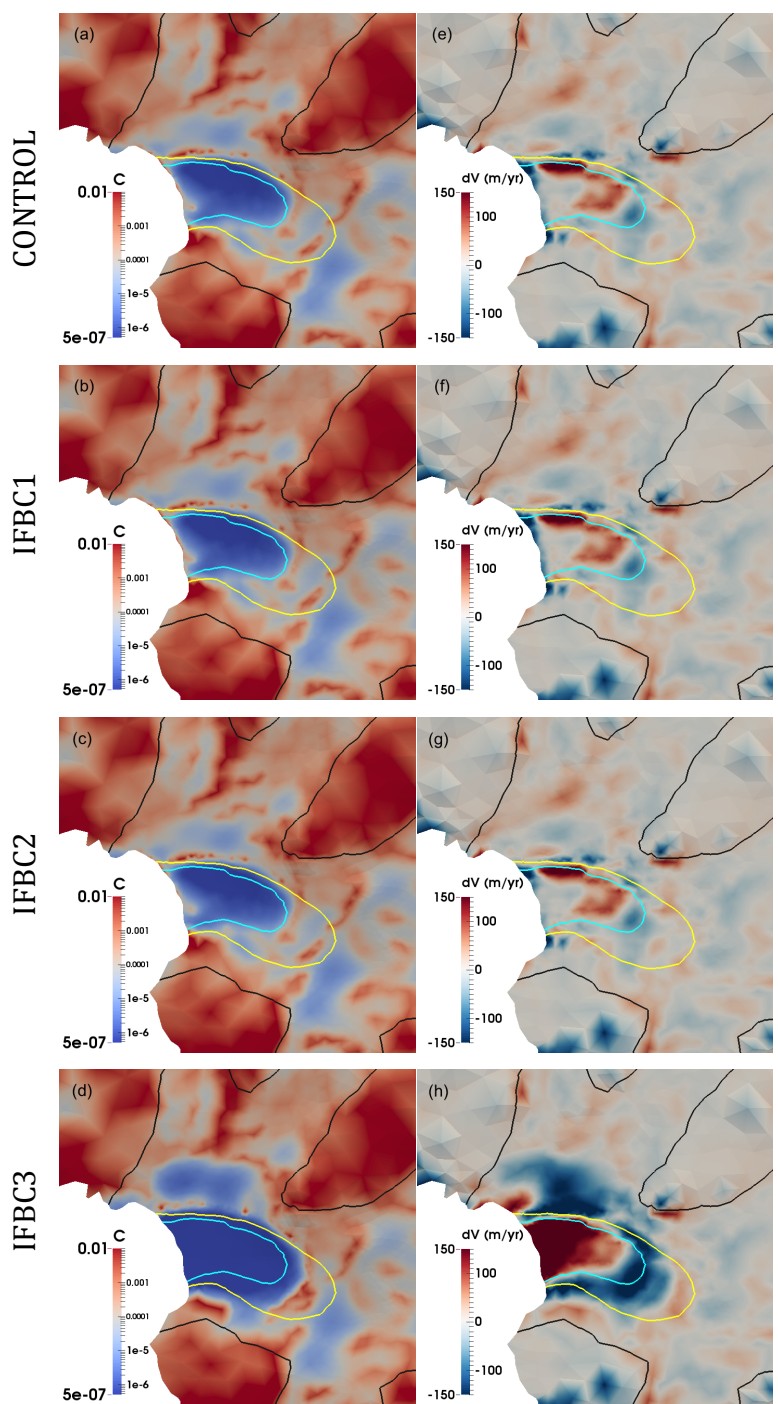

Figure 8. Distribution of basal friction coefficient C (MPa m$^{-1}$ yr) (left column), and the mismatch between the observed and modeled surface velocity (observed minus simulated; right column) from experiments: (a, e) CONTROL (first row), (b, f) IFBC1 (second row), (c, g) IFBC2 (third row), and (d, h) IFBC3 (forth row). The black, yellow, and cyan solid lines represent surface speed contours of 100 m yr$^{-1}$, 1000 m yr$^{-1}$, and 1500 m yr$^{-1}$, respectively.

805

810

Table 1. List of parameter values used in this study.

| Parameters | Symbol | Values | Units |
|---|---|---|---|
| Rheological parameter in the Arrhenius law | $A_0$ ($T < -10$ ℃) | $3.985 \times 10^{-13}$ | $Pa^{-3}\ s^{-1}$ |
| | $A_0$ ($T > -10$ ℃) | $1.916 \times 10^{3}$ | $Pa^{-3}\ s^{-1}$ |
| Activation energy in the Arrhenius law | $Q_0$ ($T < -10$ ℃) | -60 | $kJ\ mol^{-1}$ |
| | $Q_0$ ($T > -10$ ℃) | -139 | $kJ\ mol^{-1}$ |
| Gravitational constant | $g$ | 9.8 | $m\ s^{-2}$ |
| Exponent of Glen flow law | $n$ | 3 | |
| Density of ocean water | $\rho_w$ | 1025 | $kg\ m^{-3}$ |
| Density of ice | $\rho_i$ | 900 | $kg\ m^{-3}$ |

Table 2 Experiment lists. n/a is short for "not applicable". EF and SL are short for "enhancement factor" and "sea level", respectively. IF1 and IF2 represent the ice front positions located downstream and upstream of the 1996 grounding line position (Rignot et al., 2011a), respectively. RMSD is the rooted mean square deviation between the observed and simulated surface speed for the fast-flowing region of the Fleming Glacier (>1500 m yr$^{-1}$) after the third cycle.

| Experiment | EF | Bed topography used | Initial temperature in surface relaxation of Cycle 1 | Initial temperature in first inversion of Cycle 1 | SL | Ice front position | RMSD (m yr$^{-1}$) |
|---|---|---|---|---|---|---|---|
| **CONTROL** | 1.0 | bed_bm | Linear temperature | Linear temperature | 15 m | GL1996 | 75.12 |
| **TEMP1** | 1.0 | bed_bm | -20 ℃ | -20 ℃ | 15 m | GL1996 | 80.65 |
| **TEMP2** | 1.0 | bed_bm | -5 ℃ | -5 ℃ | 15 m | GL1996 | 78.07 |
| **TEMP3** | 1.0 | bed_bm | -20 ℃ | Linear temperature | 15 m | GL1996 | 78.48 |
| **EF1** | 0.5 | bed_bm | Linear temperature | Linear temperature | 15 m | GL1996 | 86.35 |
| **EF2** | 2.0 | bed_bm | Linear temperature | Linear temperature | 15 m | GL1996 | 89.38 |
| **EF3** | 4.0 | bed_bm | Linear temperature | Linear temperature | 15 m | GL1996 | 993.20 |
| **BEDZC** | 1.0 | bed_zc | Linear temperature | Linear temperature | 15 m | GL1996 | 62.60 |
| **BEDMC** | 1.0 | bed_mc | Linear temperature | Linear temperature | 15 m | GL1996 | 61.78 |
| **IFP1** | 1.0 | bed_bm | Linear temperature | Linear temperature | 15 m | IF1 | 72.10 |
| **IFP2** | 1.0 | bed_bm | Linear temperature | Linear temperature | 15 m | IF2 | 75.12 |
| **IFBC1** | 1.0 | bed_bm | Linear temperature | Linear temperature | 5 m | GL1996 | 79.38 |
| **IFBC2** | 1.0 | bed_bm | Linear temperature | Linear temperature | 25 m | GL1996 | 72.68 |
| **IFBC3** | 1.0 | bed_bm | Linear temperature | Linear temperature | n/a | GL1996 | 249.64 |