# Peer review of "Basal friction of Fleming Glacier, Antarctica, Part A: sensitivity of inversion to temperature and bedrock uncertainty"

_The Cryosphere, 2017_

## Referee Comment (RC1) · Anonymous Referee #1 · 14 Feb 2018

General comments

This paper presents results from a series of Elmer/Ice simulations of the Wordie Ice Shelf-Fleming Glacier system in West Antarctica. It aims to demonstrate the sensitivity of model inversion to englacial temperature, bedrock topography and ice front boundary, as well as provide a realistic basal shear stress field. It uses a similar multi-step inversion process to Gladstone et al. (2014), where surface relaxation is followed by an inversion for basal friction coefficient (C); then a steady-state temperature simulation using this C and velocity; and another inversion using the steady-state temperature. This process is applied iteratively in three cycles, which they show helps remove the

dependence on the initial temperature distribution. They argue this is particularly important to Fleming Glacier given the sensitivity of the system to englacial temperatures, due to the dominance of internal deformation over basal sliding. Using one of the initial temperature distributions, they run the inversion process several more times, testing the sensitivity of the inverted basal traction coefficient to bed geometry (e.g. bedmap2 versus mass-conserved), and the ice front boundary condition.

Overall this manuscript is well structured and clearly written, although some of the description of figures and discussion of results are fairly laborious and may benefit from being reduced in length. The conclusions are clearly supported by the results presented. I recommend this manuscript is published in The Cryosphere, provided the authors address the following comments.

Specific comments

Line 47: "especially for small-scale glaciers." Not sure this is relevant, or are there papers that show greater sensitivity of small- over large-scales systems?

Line 45 – 50: These two sentences appear to be contradicting each other – firstly you say that these uncertain quantities pose a challenge for modelling basal shear stress, and then you say they are not important (to that particular ice cap). I wonder if it's worth holding off on discussing the results of the Vestfonna studies until the discussion.

Line 132: Why do you make this assumption? I know it is discussed further on that the ice shelf is effectively only 1.5 km long by 2008, but before knowing this, this statement seems strange, especially given that an ice shelf is mentioned previously.

Line 163: What is your justification for using a linear sliding law?

Line 263: What do you mean by "imposed by a neigboring glacier"?

Line 274 – 277: This seems out of place here, and the related discussion in Section 4.3 is not obvious.

Line 334: add ", than CONTROL" to end of sentence? The similarity between BEDZC and BEDMC compared to CONTROL seems unsurprisingly, i.e. the two surfaces are more similar than the two thicknesses.

Line 352: Possibly worth mentioning Sun et al. (2014) here as another study that demonstrates the sensitivity of grounding line dynamics to bedrock topography.

Line 256 – 257: This seems unsurprising seeing as BEDZC makes use of surface (and mass conserved thickness?) from 2008, the same year as the velocity observations.

Line 387 – 88: Why doesn't altering the sea level affect the grounding line position?

Line 420 – 422: Not sure "spreading" is the right word: spreading in which direction?

Technical corrections

Line 21 – 23: unnecessary repetition of "temperature-dependent" deformation, combine to one sentence

Line 67: add comma at end of line

Line 108: Here FG is used for Fleming Glacier, whereas previously FGL is used. I suggest you use FG consistently (to me GL is grounding line).

Line 184: Inconsistent use of basal sliding/drag/friction coefficient, as well as inconsistent use of boldface C.

Discuss results in the present tense

Line 295: remove quotations from CONTROL"

Line 353: "most accurate", rather than "best"?

Line 403 – 410: remove quotations from simulation names, e.g. "IFBC3".

Figure 8: Could the 1500 m/yr contour be included in the other plots too, to help with comparisons?

Figure 8: Mention these are from after cycle 3

References

Sun, S., Cornford, S.L., Liu, Y., Moore, J.C., 2014. Dynamic response of Antarctic ice shelves to bedrock uncertainty. The Cryosphere 8, 1561–1576.
* * *

---

## Referee Comment (RC2) · Anonymous Referee #2 · 7 Mar 2018

**General comments**

This paper from Zhao and colleagues evaluates the sensitivity of the inversion of the basal friction coefficient of Fleming glacier, Antarctica, to (i) initial (i.e., before the inversion) temperature, (ii) different bed topographies and (iii) ice front boundary conditions. The simulations are performed with a control inverse method (MacAyeal, 1993) implemented in the Elmer/Ice ice sheet model and uses the full Stokes version of the ElmerIce model.

[Figure]

The novelty here is the use of a three-cycle spin-up (initially proposed in Gladstone et al, 2014, but for one cycle) scheme to avoid the influence of initial temperature field on the final inversion results.

The paper is quite long compared to what it could be. There is a substantial number of repetitions, which should be avoided when possible. The figures are not very clear, some differences pointed out by the authors between experiments being barely visible, thus I was not always sure by how much the three cycle methods improved the inversions results. In many places in the text I was often doubtful about the assertions. Moreover, I am not an English native speaker, but I am sure that the English could be improved. Related comments are written down below.

I have a concern with the Bedmap2 data. Since this is not written in the paper, I would like to be sure that the authors removed the difference between the Geoid and Ellipsoid height, as they did for the other DEM used, which led to have 15m of sea level. If no mistake was made with the Bedmap2 data, could you please adapt your figures to a sea level at 0, which is more common.

I question the last experiment that consists in applying different sea level at the ice front in order to deal with the uncertainties linked to the potential presence of ice mélange, the proximity of icebergs that could push back the ice stream... First, this case need to be documented with literature, or, it needs to be strongly argued. Neither the former nor the latter is done here. Another thing is that the authors have an uncertainty on the position of the ice front, I think a better experiment would be to assess the sensitivity of the results to the position of the ice front, even though I don't think that changing it by 15 km (the uncertainty) would significantly change the results.

I had issues understanding how you chose your experiments. For example, why choosing -20°C as an initial temperature pre-inversion ? Is this number related to anything real, such as a yearly average temperature ? In the paper from Schaffer 2012 that you cite, their cold and warm scenario were linked to observations, which is what you should do here, or at least explain how you chose those temperatures.

The authors need to be consistent with the terms basal drag, basal friction coefficient, basal sliding coefficient, basal shear stress. They keep mixing up those terms all over the text to mostly talking about the basal friction coefficient.

Finally, I would recommend this paper to be merged with its companion paper, also in The Cryosphere Discussion, which deals with simulating the evolution of Fleming Glacier from 2008 to 2015. All those sensitivity analysis (the first two for me) that were done in this inversion are to me verifications that you start with a sufficiently good initial state. This is not my choice of course but the one of the editor.

In all cases, this paper needs substantial rewriting before publication.

**Specific comments**

l20: I don't think you have done a sufficient number of experiment to say so, at least to say it this way. Would you explore other glaciers with the same conclusion, this assertion would be more justified.

l22: Is it true ? Looking at your fig7 I see Vb/Vs=1 over a substantial area in the ice stream part ? Means that vertical deformation here is not significant...

l24: You have done some sensitivity test, but I am not sure that those tests specifically

show the importance of what you say. I go back into this below.

l28: Here you put the glaciers of the AP and the WA ice sheet in the same category. The way those two parts of Antarctica are losing mass is fundamentally different and you should mention those differences.

l31: this sentence (mostly the same as in l14) is the kind you would find in an abstract but neither in the introduction nor in the main text.

l33: Is this always the case ? Fast flowing outlet outlet glaciers can have a small slope and be driven by basal sliding mostly, such as for the Siple coast glaciers... Could you rephrase.

l35: This way, all those processes appear to have equal impacts onto the dynamics whatever the situation...Could you rephrase. And remove strongly.

l37: Same remark as above. What is disturbing is that you seem to put all those things in the same order in influence whatever the situation.

l40: Again, this kind of sentences should be in the abstract not here, at least to me.

l42: What you infer primarily with inverse methods is basal friction (or sliding) coefficient (sometimes ice rheology). Could you rephrase.

l44: In topography, do you put basal and surface topography ? I don't think so. Maybe use the term geometry or thickness and surface topography, because we need the thickness and one of the two surfaces... Please rephrase.

l47: Why especially for small scale glacier ? We have major challenges for modelling temperature in the bigger glaciers as well. I understand you want to guide the reader to you specific case, but this comment is misleading.

l48: I feel like your analysis mostly relies on those two publications dealing with the same glacier. from that you generalise things that should not be.

l49: What type of inverse methods, did they use many ? Rephrase please.

l49: A lot of things here are not correct or need to be rephrased. 1) the results of Schafer2012 have a dependence to mesh resolution (you should read section 4.3). 2) this is not as simple as that for bed topography and velocity uncertainties. You should be less approximative in your assertions.

l51: This sentence is not clear, rephrase please.

l52: And I don't think you are doing this generalisation in your paper. This is clearly overstating to me.

l54: Do you test this to all the inversion methods. please rephrase.

l56: What robust means here ? You will have tested on one single friction law, and almost the simplest one. You should rephrase.

l60: Maybe here you could add some figures, what are the velocities, the size, some more details about the glacier...

l65: You invert the basal friction (or sliding) coefficient. You need to be consistent over the text.

l66: What you invert is the basal friction coefficient. Rephrase please.

l80: Just a question here to be sure because you don't mention it after. Did you make sure you accounted for the Geoid-Ellipsoide difference for Bedmap2, which reference is the Geoid ?

l82: This is rather strange and unusual to use sea level of 15m. It would be much clearer to take the geoid as the reference.

l86: Since you mentioned the Wordie ice shelf in the previous section, you should replace "This"

l87: shear stress $->$ friction coefficient

l95: Could you break down this sentence in two parts, otherwise this is hard to read.

l100: To calculate the Hmc, did you use ElmerIce ? I think it needs to be mentioned since this would not be an official feature in Elmer.

l103: This is not really true for bedzc since Sbm has a resolution of 1000m. How did you interpolate Sbm from 1000m to 500m ?

l107: could you mention the fact that they are both part of the same basin.

l112: shear stress $->$ friction coefficient

l124: basal drag $->$ basal friction coefficient

l134: Here you need to mention the difference that you have between your reconstructed ice front and the grounding line of Rignot2011a.

l144: My personal viewpoint is that the mesh resolution influence should always be checked beforehand... This is not such a strenuous task to do this.

l149: The temperature is fixed to what dataset ?

l152: You describe the BC and then you switch into something different, which should be more in the discussion section, not here. This way of writing just affect the reading in a bad way. Please consider not doing this in the text.

l159: Temporarily : what does it mean ?

l169: Ah here you talk about temperature data. It should be written in the same place as above.

l178: Ok, Why 0.2 ? Did you check other values ?

l186: drag $->$ sliding

l187: As there are many types of cost functions in the literature, you should define yours.

l193: Here I think you should cite Gillet2012 as it seems that you do exactly the same thing for the cost function

l200: You should add a figure showing the improvement made with E=2.5. I would also be very pleased to see the L-curve, for instance in a supplementary.

l207: Actuality: I am not sure we can use this word here, change please

l210: If you say so, you need to show that Greenland glaciers and the domain of your study can be similar to each other. Or you need to rephrase your sentence...

l215: you mention Gong2016 (this is 2017 actually) for the spin up scheme or for Elmerice. For the latter, better to cite Gagliardini2013

l219: There is a step here that is not common, surface relaxation with C at its initial chosen value. What is done usually is the inversion, then the relaxation over about 15 years. I wonder the effect of the surface relaxation using a C that is far from reality...

l220: Basal sliding

l225: Means you don't account for the modification of surface with relaxation at the beginning of the last two cycles ?

l228: Basal friction

l229: To your inverse method, not all of them

l243: Don't say linear but rather Control

l246 to l265: I don't really understand the relevancy of this scenario. To me you should rather study the influence of the position of your ice front, since this is what you are not sure about with your hypothesis assuming ice front = grounding line.

l267: Results and discussion

l270: what do you call robustness here ? Replace drag by sliding. Rephrase please.

l273: There are only 3 TEMP experiments, be more clear

l275: Here what we need to have is a metric like the RMS, otherwise this is only a maximum value that is not representative of the rest of the data.

l277: I don't understand what you say here ?

l279: This is quite difficult to evaluate the differences between the different experiments in your maps. I would recommend to the relative differences with a reference experiment.

l281: Figures should be ordered differently, such vertically Control, Temp1, Temp2, Temp3, this is otherwise very difficult to follow.

l283: Looking at Schaffer2012, it does not seem to me that the dependence of their model to temperature scenario is smaller than yours... You do need to quantify your differences, because this is really not clear.

l286: They showed a non influence onto the modelled surface velocity, not the friction coefficient, or I misread their paper... Their Fig8 shows the differences in terms of basal friction coefficient, but this slightly affect surface velocity as the inverse model tends to minimize the differences.

l289 to l291: Already said, please avoid repetitions. You are in the result and discussion, thus adding other unnecessary stuff is only distracting the reader.

l283: I think this is normal to have different results if you choose a sort of outlier in your initial state, like -20 degrees everywhere for the initial state. I don't think you discussed this as a comparison with the final result ? Is -20 in the range of this final result ?

l291: Drag $->$ friction coefficient

l295: Could you quantify your sticky spots ?

l296: "therefore..." remove this as this was already written
l300: You should say Control instead of linear scenario

l306 to l308: Third time I see this in the paper, remove repetitions please.

l306: The low impact is on modelled surface velocity. There is an impact on basal friction coefficient (or basal drag as they say)

l313: No need to say "inside the yellow contour" in the text

l318: "shows that internal deformation": you should vertical deformation here.

l319: I don't agree with this assertion. Vb=Vs in the fastest flowing areas. In between those you have an area with Vb much lower than Vs, but this matches the places where driving stress is much higher. So this is the driven stress that may drive the vertical deformation, not only the ice internal temperature... You need to rephrase.

l330 to l332: not necessary because already mentioned

l334: Remove mentions to colors and rather explain with the physical parameters

l340: I don't understand

l345 to 347: Why mentioning the MISI in a paper that only deals with inversions, there is no point to me.

l347: basal friction

l350: What is behind "it" ? The link with previous sentences is not quite clear. Rephrase please.

l355: Ok great, you calculated RMSEs. However, 1) you should have done it before (see previous comment above) and 2) please give us numbers.

l357: I guess this is justified by your RMSE. I think you should discuss more this result, as it suggest that using data taken over a short time range improve the results compared to Bedmap2, which is taken over larger time scales than a year... If I am not wrong.

l368 to l371: remove this as already written in the methods section

l371: You did not really have investigated the sensitivity to uncertainty to me. You only have tested two datasets, one being more accurate than the other by the way. The Mass conservation based inversion for bedrock is quite an efficient method to infer the bedrock (see Morlighem2014 NG)

l382: This is the kind of things you need to check really. You may have the answer in the paper by Mouginot 2012 in the Journal Remote Sensing. It seems to be a combination between 2007 to 2009 data.

l387: I really question the relevancy of this experiment. Why doing so as it seems to me that more relevant experiment would be to adjust the ice front position, where you have your uncertainty, and check the sensitivity of inversion results. This latter experiment would not change mush the results to me, because over 1.5 km of ice shelf, you don't have much buttressing, but it would be more relevant than what you propose to me.

l421: I don't understand, in what context ?

l429: This is still about this experiment. To test such an amplitude in the influence of sea levels in inversion results, you need to cite literature about what buttressing could be added from ice mélange (see Krug2014 by the way)...

Figure 4: add relaxation time here

Figure 5 caption: Temp4 doesn't exist

Figures in general: All the differences that you comment are not always visible. These are to me really tight differences so if you want to argue on this to underline the improvement that are brought by your 4 cycle spin up scheme, you should care more about the figures. Use relative differences between the Control and the other experiments.

Figures in general: please, for the readability order vertically your subplots like: Control,

temp1, temp2, temp3

Figure 7: Here is certainly a way to remove those zigzags discontinuity, I know Paraview is not user friendly for some stuff, but I don't think this is acceptable for a peer reviewed paper.

---

## Author Response (AR1)

We are grateful to Reviewer 1 for the positive and constructive suggestions to improve our paper. We have addressed all comments below. The line numbers in the responses are based on the revised manuscript without track changes.

Please note that Mathieu Morlighem created the ice thickness data for the Fleming Glacier system using mass conservation method, which is very important for most experiments done in this study. We do value his contribution to this paper, so we add him as the co-author in the revised text.

In response to the reviewer 2's question about our choice of enhancement factor, we implemented a new sensitivity test. This was more thorough than our original test, and with a more up-to-date setup. And in fact, it reveals that our original choice was not optimal. So we added the sensitivity tests for various E values (0.5, 1.0, 2.0, 4.0) in Sect. 3.6 and Sect. 4.2, and the optimal value of 1.0 was chosen as the E in the CONTROL experiment. We redid all the simulations and modified the text and figures accordingly. Our conclusions have not changed.

General comments

This paper presents results from a series of Elmer/Ice simulations of the Wordie Ice Shelf-Fleming Glacier system in West Antarctica. It aims to demonstrate the sensitivity of model inversion to englacial temperature, bedrock topography and ice front boundary, as well as provide a realistic basal shear stress field. It uses a similar multi-step inversion process to Gladstone et al. (2014), where surface relaxation is followed by an inversion for basal friction coefficient (C); then a steady-state temperature simulation using this C and velocity; and another inversion using the steady-state temperature. This process is applied iteratively in three cycles, which they show helps remove the dependence on the initial temperature distribution. They argue this is particularly important to Fleming Glacier given the sensitivity of the system to englacial temperatures, due to the dominance of internal deformation over basal sliding. Using one of the initial temperature distributions, they run the inversion process several more times, testing the sensitivity of the inverted basal traction coefficient to bed geometry (e.g. bedmap2 versus mass-conserved), and the ice front boundary condition.

Overall this manuscript is well structured and clearly written, although some of the description of figures and discussion of results are fairly laborious and may benefit from being reduced in length. The conclusions are clearly supported by the results presented. I recommend this manuscript is published in The Cryosphere, provided the authors address the following comments.

Specific comments

Line 47: "especially for small-scale glaciers." Not sure this is relevant, or are there

papers that show greater sensitivity of small- over large-scales systems?

To our knowledge, no study has shown that. We removed "especially for small-scale glaciers".

Line 45 – 50: These two sentences appear to be contradicting each other – firstly you say that these uncertain quantities pose a challenge for modelling basal shear stress, and then you say they are not important (to that particular ice cap). I wonder if it's worth holding off on discussing the results of the Vestfonna studies until the discussion.

The first sentence is a general statement for most glaciers, which we quote Vaughan and Arthern (2007). But, the Vestfonna Ice Cap is mentioned as a case showing the less sensitivity to the basal topography, which is contrasting to what we find for the Fleming Glacier in this study. The Fleming Glacier is the main focus of this paper, but we think it is good to mention the Vestfonna here.

Line 132: Why do you make this assumption? I know it is discussed further on that the ice shelf is effectively only 1.5 km long by 2008, but before knowing this, this statement seems strange, especially given that an ice shelf is mentioned previously.

We did not have a clear way to provide the ice thickness for a short fringing ice shelf left, which is detected from the DEM data in Jan 2008 (we clarify this in Sect. 2.1 and Fig. S1). This small ice shelf disappeared in Apr 2008, as shown in Fig. 1c. To discuss the sensitivity to different ice front position, we expanded remarks in Sect. 3.6 and Sect. 4.3, and relevant results and discussions have been added to the text.

Line 163: What is your justification for using a linear sliding law?

Different sliding laws in inverse modeling will not change the inversed basal shear stress distribution, and it will just lead to different basal friction coefficients based on different sliding law. In diagnostic studies that invert to find the basal shear stress which gives the best agreement with observed surface velocities, the choice of sliding "law" is not relevant provided that the required stress can be generated by adjustments of the parameters in the sliding law – in this case the coefficient C. The inversion procedure modifies C to modify stress – adjusting the momentum balance. That solution of the Stokes equation provides an updated estimate of basal velocity – which enters the next cycle of the inversion search. The question does remain whether this is physically suitable relationship to apply when the system is evolving, but this is not relevant here. So we adopted the simplest sliding law here following Gagliardini et al. (2013); Gillet-Chaulet et al. (2012). We clarified this in the text (Line 190).

Line 263: What do you mean by "imposed by a neigboring glacier"?

We made a hypothesis that the ice front of the Fleming Glacier had a continuation of the advancing glacier by exerting a normal stress on the ice front. Here we modified into "imposed by a hypothetical undeforming continuation of the advancing glacier".

Line 274 – 277: This seems out of place here, and the related discussion in Section 4.3 is not obvious.

Now that we have adopted the E of 1.0 as the CONTROL setup, we find that the surface lowering near the ice front during the surface relaxation was <25 m in each cycle. But we still need to know whether the small changes in surface elevation at the ice front will affect the basal friction deduced from inversion, which is discussed in Sect. 4.4. So we modified this sentence to a separate paragraph and modified the

relevant discussions in Sect. 4.4 (Line 333-337).

Line 334: add ", than CONTROL" to end of sentence? The similarity between BEDZC and BEDMC compared to CONTROL seems unsurprisingly, i.e. the two surfaces are more similar than the two thicknesses.

We added ", compared with CONTROL".

Line 352: Possibly worth mentioning Sun et al. (2014) here as another study that demonstrates the sensitivity of grounding line dynamics to bedrock topography.

Added.

Line 356 – 357: This seems unsurprising seeing as BEDZC makes use of surface (and mass conserved thickness?) from 2008, the same year as the velocity observations.

Yes, we agree. To clarify this, we clarified this in Line 438-440) "Both BEDMC and BEDZC use the 2008 surface DEM and this improvement over the Bedmap2 surface DEM in CONTROL appears significant, even before turning to the matter of ice thickness. "

Line 387 – 88: Why doesn't altering the sea level affect the grounding line position?

We ran all the experiments with the grounding line fixed. The sea level adjustments are meant as a convenient tool for altering the force applied at the ice front, including the influence of uncertainties in ice thickness (and hence bed depth) at the ice front/grounding line. We have clarified this in Line 302-304.

Note that the height above buoyancy calculations for 2008 in the companion paper (Zhao et al., companion paper) indicate that the glacier – as described by our datasets – would have remained grounded at the ice front for all but the largest sea level forcing.

Line 420 – 422: Not sure "spreading" is the right word: spreading in which direction?

By "spreading" we meant longitudinal extensional flow. We modified this sentence into "The lowered surface at the ice front in experiments IFBC1 and CONTROL is apparently the consequence of rapid deformation due to its own weight (longitudinal extension with locally high vertical shear) of an ice cliff, which is over 100 m higher than the control sea level. " (Line 526-529).

Technical corrections

Line 21 – 23: unnecessary repetition of "temperature-dependent" deformation, combine to one sentence

Modified.

Line 67: add comma at end of line

Added.

Line 108: Here FG is used for Fleming Glacier, whereas previously FGL is used. I suggest you use FG consistently (to me GL is grounding line).

Modified "FGL" into "FG" in whole text.

Line 184: Inconsistent use of basal sliding/drag/friction coefficient, as well as inconsistent use of boldface C. Discuss results in the present tense

Modified all these terms "basal sliding/drag/friction coefficient" into "basal friction

coefficient". The font in equations is unchangeable so we could just make sure all the $C$ in the main text shares the same font. We have adjusted the tenses used in the paper for consistency.

Line 295: remove quotations from CONTROL"

Deleted

Line 353: "most accurate", rather than "best"?

Modified into "more accurate"

Line 403 – 410: remove quotations from simulation names, e.g. "IFBC3".

Deleted

Figure 8: Could the 1500 m/yr contour be included in the other plots too, to help with comparisons?

Added

**References**

Gagliardini, O., Zwinger, T., Gillet-Chaulet, F., Durand, G., Favier, L., de Fleurian, B., Greve, R., Malinen, M., Martín, C., Råback, P., Ruokolainen, J., Sacchettini, M., Schäfer, M., Seddik, H., and Thies, J.: Capabilities and performance of Elmer/Ice, a new-generation ice sheet model, Geosci. Model Dev., 6, 1299-1318, 2013.

Gillet-Chaulet, F., Gagliardini, O., Seddik, H., Nodet, M., Durand, G., Ritz, C., Zwinger, T., Greve, R., and Vaughan, D. G.: Greenland ice sheet contribution to sea-level rise from a new-generation ice-sheet model, The Cryosphere, 6, 1561-1576, 2012.

Vaughan, D. G. and Arthern, R.: Why Is It Hard to Predict the Future of Ice Sheets?, Science, 315, 1503-1504, 2007.

Zhao, C., Gladstone, R., Zwinger, T., Warner, R., and King, M. A.: Basal friction of Fleming Glacier, Antarctica, Part B: implications of evolution from 2008 to 2015, The Cryosphere, companion paper. companion paper.

Response to the Interactive comment on

"Basal drag of Fleming Glacier, Antarctica, Part A: sensitivity of inversion to temperature and bedrock uncertainty"

by Chen Zhao et al.

Anonymous Referee #2

We are grateful to reviewer 2 for the positive and constructive suggestions to improve our paper. In particular, we now explore the effect of moving the location of the ice front, as suggested. We have addressed the comments below. The line numbers in the responses are based on the revised manuscript without track changes.

Please note that Mathieu Morlighem created the ice thickness data for the Fleming Glacier system using mass conservation method, which is very important for most experiments done in this study. We do value his contribution to this paper, so we add him as the co-author in the revised text.

In response to the question about our choice of enhancement factor, we implemented a new sensitivity test to enhancement factor (E). This was more thorough than our original test, and with a more up-to-date setup. In fact it reveals that our original choice was not optimal. We added the sensitivity tests to various E values (0.5, 1.0, 2.0, 4.0) as described in Sect. 3.6 and discussed in Sect. 4.2. The optimal value E = 1.0 was chosen as the enhancement factor in all the other experiments. We redid all the simulations and modified the text and figures. We retain the sensitivity tests for the multi-cycle inversion scheme as the first results presented, since in all other cases only the third cycle results are discussed. Our conclusions have not changed.

General comments

This paper from Zhao and colleagues evaluates the sensitivity of the inversion of the basal friction coefficient of Fleming glacier, Antarctica, to (i) initial (i.e., before the inversion) temperature, (ii) different bed topographies and (iii) ice front boundary conditions. The simulations are performed with a control inverse method (MacAyeal, 1993) implemented in the Elmer/Ice ice sheet model and uses the full Stokes version of the Elmer/Ice model.

The novelty here is the use of a three-cycle spin-up (initially proposed in Gladstone et al, 2014, but for one cycle) scheme to avoid the influence of initial temperature field on the final inversion results.

The paper is quite long compared to what it could be. There is a substantial number of repetitions, which should be avoided when possible. The figures are not very clear, some differences pointed out by the authors between experiments being barely visible, thus I was not always sure by how much the three cycle methods improved the inversions results. In many places in the text I was often doubtful about the assertions. Moreover, I am not an English native speaker, but I am sure that the English could be improved. Related comments are written down below.

I have a concern with the Bedmap2 data. Since this is not written in the paper, I would

like to be sure that the authors removed the difference between the Geoid and Ellipsoid height, as they did for the other DEM used, which led to have 15m of sea level. If no mistake was made with the Bedmap2 data, could you please adapt your figures to a sea level at 0, which is more common?

All data used in the study are self-consistent which is the key concern here. In this study we adopted an ellipsoidal height references for all datasets (surface and bedrock elevation data) (WGS84 ellipsoid). To clarify this, we added a few words in Sect. 2.2 (Line 96-98) "The first is from the Bedmap2 dataset (Fretwell et al., 2013), with a resolution of 1 km (hereafter bed_bm; Fig. 2b), which is converted from the EIGEN-GL04C geoid to WGS84 ellipsoid heights. "

We don't think the issue of the reference value of sea level should cause confusion and we are sure all the elevation data is under the same height reference system. To be quite clear – the 15 m sea level elevation is determined from examining the 2008DEM used in the paper for the difference between elevations over the ocean and the glacier. But we agree to adapt my figures (Fig. 2b and Figs. 7g-i) to the meters above sea level with a sea level at 0 m.

I question the last experiment that consists in applying different sea level at the ice front in order to deal with the uncertainties linked to the potential presence of ice mélange, the proximity of icebergs that could push back the ice stream... First, this case need to be documented with literature, or, it needs to be strongly argued. Neither the former nor the latter is done here.

The mélange issue is not the main or only reason for exploring different force balances at the ice front – as stated in Sect. 3.6 (Line 301-305). Uncertainties in ice thickness/bed elevation are also a major consideration. The emergence of a curious sticky spot with high basal friction adjacent to the ice front further encouraged these sensitivity tests. Many previous studies have also argued that the ice mélange could suppress calving by exerting a buttressing force directly on the glacier terminus (Amundson et al., 2010; Krug et al., 2015; Robel, 2017; Todd and Christoffersen, 2014; Walter et al., 2017). We have added this in the main text (Sect. 3.6, Line 300).

Another thing is that the authors have an uncertainty on the position of the ice front, I think a better experiment would be to assess the sensitivity of the results to the position of the ice front, even though I don't think that changing it by 1.5 km (the uncertainty) would significantly change the results.

Thanks for this good point. We additionally conducted sensitivity tests to three different ice front positions in Sect. 3.6. It did not make a significant change, as expected by the reviewer, but different ice front positions affected the basal friction near the ice front. Relevant results and discussions have been added in Sect. 4.4.

I had issues understanding how you chose your experiments. For example, why choosing -20 C as an initial temperature pre-inversion? Is this number related to anything real, such as a yearly average temperature? In the paper from Schaffer 2012 that you cite, their cold and warm scenario were linked to observations, which is what you should do here, or at least explain how you chose those temperatures.

We don't have any observations for the temperature field except for the surface temperature from RACMO model, which ranges from -26 C to -7 C. The choice of -20 C or -5 C as an initial englacial temperature is not based on observations. In the Glen Flow law, the ice temperature is a function of pressure melting point via the Arrhenius law (Gillet-Chaulet et al., 2012):

$$A = A_0 e^{(-Q/[R(273.15+T)])}$$

Here, $A_0$ is the pre-exponential factor and $Q$ is activation energy. $A_0$ and $Q$ have different values while the temperature T is lower or higher than -10 C. To test the sensitivity of inverse methods to the initial englacial temperature, we assumed two constant values, one is lower than -10 C and the other one higher.

The authors need to be consistent with the terms basal drag, basal friction coefficient, basal sliding coefficient, basal shear stress. They keep mixing up those terms all over the text to mostly talking about the basal friction coefficient.

Thanks for pointing this out. We have changed all the terms to use "basal friction coefficient".

Finally, I would recommend this paper to be merged with its companion paper, also in The Cryosphere Discussion, which deals with simulating the evolution of Fleming Glacier from 2008 to 2015. All those sensitivity analysis (the first two for me) that were done in this inversion are to me verifications that you start with a sufficiently good initial state. This is not my choice of course but the one of the editor.

This paper proposed the multi-cycle spin-up scheme to remove the effect of the plausible initial temperature assumption for the glaciers like the Fleming Glacier, which have strong, temperature-dependent, deformational flow in the fast-flowing regions. Sensitivity tests to various bedrock datasets and ice front boundary conditions for the Fleming system provided a good initial state and setting up for further simulations on this system. If we combine this paper with its companion paper, most of the above points would have to be put into the supplementary sections, which is not good for benefiting more researchers interested in the technical spin-up aspect. So we prefer keeping the two papers separate. In particular, with the addition of the ice front position sensitivity tests suggested by the reviewer this paper contains quite sufficient material to stand alone.

In all cases, this paper needs substantial rewriting before publication.

Specific comments

l20: I don't think you have done a sufficient number of experiment to say so, at least to say it this way. Would you explore other glaciers with the same conclusion, this assertion would be more justified.

We gave this conclusion for glaciers like the Fleming system. To clarify this, we combined this sentence and next sentence into "This is particularly important for glaciers like the Fleming Glacier, which have areas of strongly temperature-dependent, deformational flow in the fast-flowing regions ". We also modified "three cycle" into "multi-cycle" (Line 23-25).

l22: Is it true ? Looking at your fig7 I see Vb/Vs=1 over a substantial area in the ice stream part ? Means that vertical deformation here is not significant...

Looking at Fig. S5b, there is a steep region between the 1000 m yr$^{-1}$ and 1500 m yr$^{-1}$, where Vb is much smaller than the Vs. It means that the vertical deformation in the some parts of the fast flowing regions is significant.

l24: You have done some sensitivity test, but I am not sure that those tests specifically show the importance of what you say. I go back into this below.

We respond to this later at the relevant point.

l28: Here you put the glaciers of the AP and the WA ice sheet in the same category. The way those two parts of Antarctica are losing mass is fundamentally different and you should mention those differences.

We are aware that the ice shelf collapse in the AP is likely significantly driven by surface melting, and the ice shelves in the AP are more vulnerable to atmospheric warming. However, the Fleming Glacier in this study had nearly lost its ice shelf (the Wordie Ice Shelf) by 2008.

In recent studies on the Fleming Glacier (Friedl et al., 2018; Walker and Gardner, 2017), it is proposed that the glacier acceleration and thinning is likely to be triggered by the incursion of warm ocean water, associated with grounding line retreat, which has shown the possibility that some glaciers of the AP may lose mass in the same way with those in the WA.

We agree with the reviewer that it is important to consider both the similarity and difference between these regions, and we do extensively discuss this in our companion paper (Zhao et al., companion paper).

l31: this sentence (mostly the same as in l14) is the kind you would find in an abstract but neither in the introduction nor in the main text.

We think this comment is a personal preference rather than a scientific critical argument. If the reviewer wishes to give a reason why it is not appropriate to put this sentence in the Introduction, we would consider removing it. Regarding the apparent duplication, our view is that an Abstract is a summary, not a substitute for aspects of the Introduction

l33: Is this always the case ? Fast flowing outlet glaciers can have a small slope and be driven by basal sliding mostly, such as for the Siple coast glaciers... Could you rephrase.

We modified this sentence into "The high velocities of fast-flowing outlet glaciers arise from internal ice deformation or ice sliding at the bed or both. " (Line 35-36).

l35: This way, all those processes appear to have equal impacts onto the dynamics whatever the situation...Could you rephrase. And remove strongly.

We simply listed all the relevant factors regarding deformation here and we are not emphasizing the importance of each impact. We are happy to remove "strongly" since we do not discuss relative importance.

l37: Same remark as above. What is disturbing is that you seem to put all those things in the same order in influence whatever the situation.

Same response as to l35.

l40: Again, this kind of sentences should be in the abstract not here, at least to me.

Same response as to l31.

l42: What you infer primarily with inverse methods is basal friction (or sliding) coefficient (sometimes ice rheology). Could you rephrase.

Modified "basal shear stress" to "basal friction coefficients", added "ice rheology". An inversion could produce basal velocities but it deduces basal shear stress by adjusting the basal friction coefficient in the description of basal shear stress inside a sliding law as a boundary condition to solving the momentum balance equations. So we don't agree that the basal shear stress is not the target of the inverse approach here.

l44: In topography, do you put basal and surface topography ? I don't think so. Maybe use the term geometry or thickness and surface topography, because we need the thickness and one of the two surfaces... Please rephrase.

Modified "glacier topography" into "glacier geometry"

l47: Why especially for small scale glacier ? We have major challenges for modeling temperature in the bigger glaciers as well. I understand you want to guide the reader to you specific case, but this comment is misleading.

Thanks for the reviewer's suggestion. We deleted "especially for small-scale glaciers".

l48: I feel like your analysis mostly relies on those two publications dealing with the same glacier. from that you generalise things that should not be.

It is not our intention to generalize the Vestfonna case here. The Fleming case turns out to be a contrasting one. We are happy to address it further if there are specific concerns about it.

l49: What type of inverse methods, did they use many ? Rephrase please.

We unintentionally suggested they used a range of techniques – they used the "Robin inverse method". We corrected this in the text (Line 51).

l49: A lot of things here are not correct or need to be rephrased. 1) the results of Schafer2012 have a dependence to mesh resolution (you should read section 4.3). 2) this is not as simple as that for bed topography and velocity uncertainties. You should be less approximative in your assertions.

1) Thanks for pointing out this. Yes, the results of Schafer 2012 emphasized the importance of a finer mesh. So we delete "mesh resolution or".

2) The Sect. 4.4 of Schafer 2012 did show that the inverse method is not sensitive to the modification of the surface and bed elevation datasets.

l51: This sentence is not clear, rephrase please.

Modified into "In their case, sliding dominated the flow regime, and the impact of internal deformation on ice velocity was relatively small compared to the important role of friction heating at the bed on the basal sliding " (Line 52-54).

l52: And I don't think you are doing this generalisation in your paper. This is clearly overstating to me.

We just state that "No generalization on these findings to Antarctic outlet glaciers has been investigated", but we did not mean to do this generalization in this paper. To make it clearer, we changed this sentence into "It is unclear whether this property is specific to Vestfonna situation or if it also applies to other fast flowing glaciers." (Line 55).

l54: Do you test this to all the inversion methods. please rephrase.

Modified into "to test the sensitivity of a variational inverse method (MacAyeal, 1993; Morlighem et al., 2010) for basal friction to basal geometry and to an assumed initial englacial temperature distribution for a different outlet glacier system" (Line 56-59).

l56: What robust means here ? You will have tested on one single friction law, and almost the simplest one. You should rephrase.

"Robust" here means the robustness of simulated basal friction coefficient distribution to experiment design and the mismatch between the simulated and observed surface velocities. We don't want our simulated results to be dependent on our initial temperature assumptions. As discussed in the response to Reviewer 1, in diagnostic studies of the type we present here, the claimed physical character of the basal friction law is of little importance (assuming that it can produce the required range of basal shear stresses) so reliance on a single friction law is not a limitation. So we think it is appropriate to use "robust" here.

l60: Maybe here you could add some figures, what are the velocities, the size, some more details about the glacier...

We added a sentence (Line 63-68) "The Fleming Glacier (FG) (Fig. 1b), as the main tributary glacier, has a current length of ~80 km and is ~10 km wide near the ice front (Friedl et al., 2018). This glacier has recently shown a rapid increase in surface-lowering rates (doubling near the ice front after 2008) (Zhao et al., 2017), and the largest velocity changes (> 500 m yr$^{-1}$ near the ice front) across the whole Antarctic over 2008-2015 (Walker and Gardner, 2017). "

l65: You invert the basal friction (or sliding) coefficient. You need to be consistent over the text.

Modified for whole text.

l66: What you invert is the basal friction coefficient. Rephrase please.

Modified.

l80: Just a question here to be sure because you don't mention it after. Did you make sure you accounted for the Geoid-Ellipsoide difference for Bedmap2, which reference is the Geoid ?

Yes, we adopted the bedmap2 data based on the WGS84 ellipsoid and we clarified this in Sect. 2.2 (Line 96-98). "The first is from the Bedmap2 dataset (Fretwell et al., 2013) with a resolution of 1 km (hereafter bed_bm; Fig. 2b), which is converted from the EIGEN-GL04C geoid to WGS84 ellipsoid heights. " See also the discussion above under response to General Comments.

l82: This is rather strange and unusual to use sea level of 15m. It would be much clearer to take the geoid as the reference.

As we stated above, the value of sea level will not make a difference in our experiments as long as we are sure all the elevation data is under the same height reference system. To be quite clear – the 15 m sea level elevation is determined from examining the 2008DEM used in the paper for the difference between elevations over the ocean and the glacier. But we agree to adapt my figures (Fig. 2b and Figs. 7g-i) to the meters above sea level with a sea level at 0 m.

l86: Since you mentioned the Wordie ice shelf in the previous section, you should replace "This"

"this region" -> "the WIS-FG system"

l87: shear stress - > friction coefficient

Modified.

l95: Could you break down this sentence in two parts, otherwise this is hard to read.

Modified.

l100: To calculate the Hmc, did you use ElmerIce ? I think it needs to be mentioned since this would not be an official feature in Elmer.

No, we calculated Hmc using ISSM's mass conservation algorithm (Morlighem et al. 2011). We clarified the manuscript accordingly (Line 105-111) "$H_{mc}$ (where "mc" refers to "mass conservation") is the ice thickness data with a resolution of 450 m covering three regions shown in Fig. 2e. $H_{mc}$ for the yellow area is computed using the Ice Sheet System Model's mass conservation method (Morlighem et al., 2011; Morlighem et al., 2013), based on ice thickness measurements from the Center for Remote Sensing of Ice Sheets (CReSIS), using ice surface velocities in 2008 from Rignot et al. (2011b), surface accumulation from RACMO 2.3 (van Wessem et al., 2016) and 2002-2008 ice thinning rates from Zhao et al. (2017). The thickness data for the grey area is interpolated from Bedmap2 (Fretwell et al., 2013), while the data in the red area ensures a smooth transition between the two regions. The yellow area indicates the Fleming Glacier system with ice velocity >100 m yr$^{-1}$. ".

l103: This is not really true for bedzc since Sbm has a resolution of 1000m. How did you interpolate Sbm from 1000m to 500m ?

We presume you meant to talk about bed_mc here. We used a bilinear interpolation to downscale Sbm to 500 m. We have clarified this in the manuscript (Line 103).

l107: could you mention the fact that they are both part of the same basin.

Whether or not they are in the same "basin" depends on one's precise definition of a basin. What we mean is that each of these features has its own local minimum in bedrock elevation and a significant region of reverse bed slope. We have modified the text to make it clearer to the reader that both features are under the Fleming main trunk (Line 121-123).

l112: shear stress - > friction coefficient

Modified.

l124: basal drag - > basal friction coefficient

Modified.

l134: Here you need to mention the difference that you have between your reconstructed ice front and the grounding line of Rignot2011a.

Here we mentioned that the ice front position in 2008 was assumed to be same with the 1996 grounding line of Rignot et al. (2011a). So there is no difference here.

l144: My personal viewpoint is that the mesh resolution influence should always be checked beforehand... This is not such a strenuous task to do this.

Another experiment has been done with 20 vertical layers. The simulated C shows nearly the same distribution as the CONTROL experiment. So we modified this sentence into "In the current study an experiment with 20 extruded layers (not shown) gives very similar results as with 10 layers, confirming those findings also apply to the WIS-FG system. " (Line 164-165).

l149: The temperature is fixed to what dataset ?

The surface temperature is fixed to the yearly average surface temperature over 1979-2014 computed from RACMO2.3/ANT27. We have moved the relevant paragraph

after this sentence (Line 173-179).

l152: You describe the BC and then you switch into something different, which should be more in the discussion section, not here. This way of writing just affect the reading in a bad way. Please consider not doing this in the text.

Thanks for your suggestion. We deleted this sentence. The uncertainties of ice thickness and bedrock topography, the low accuracy of ice front and grounding line locations, and the possible buttressing on the ice front by partly detached icebergs and ice mélange are now discussed in Sect. 3.6 and Sect. 4.4.

l159: Temporarily : what does it mean ?

Thank you for the query. We meant "temporally fixed" and have corrected accordingly.

l169: Ah here you talk about temperature data. It should be written in the same place as above.

This whole paragraph has been moved to the Line 173.

l178: Ok, Why 0.2 ? Did you check other values ?

Yes, we checked longer time and shorter times. Shorter time was not enough for Elmer/Ice to remove the non-physical spikes, which would lower the efficiency of following inverse running. If we relaxed the free surface for longer than 0.2 yr, the relaxed surface was much lower or higher than the observed one, since the simulated velocity close to the front was very high.

l186: drag - > sliding

"drag" -> "friction"

l187: As there are many types of cost functions in the literature, you should define yours.

Added.

l193: Here I think you should cite Gillet2012 as it seems that you do exactly the same thing for the cost function

Added "(following for example Gillet-Chaulet et al. (2012))"

l200: You should add a figure showing the improvement made with E=2.5. I would also be very pleased to see the L-curve, for instance in a supplementary.

Thanks for your suggestions. The L-curve analysis figure has been added as Fig. S2 in the supplementary material.

As we mentioned above, we implemented a new sensitivity test to the enhancement factor E. This was more thorough than our original informal test, and with a more up-to-date setup. And in fact it reveals that our original choice was not optimal. So we added the sensitivity tests to various E values (0.5, 1.0, 2.0, 4.0) in Sect. 3.6 and Sect. 4.2, and the optimal value of 1.0 was chosen as the E value in the CONTROL experiment. We redid all the simulations and modified relative text and figures as required.

l207: Actuality: I am not sure we can use this word here, change please

"Actuality" -> "Reality"

l210: If you say so, you need to show that Greenland glaciers and the domain of your study can be similar to each other. Or you need to rephrase your sentence...

We guess you refer to l209 in the original text? We delete "However" for a subtle shift of emphasis. The current temperature distribution in the Fleming Glacier cannot be accurately calculated or estimated in any way. Steady state is as good a guess as anything else.

l215: you mention Gong2016 (this is 2017 actually) for the spin up scheme or for Elmerice. For the latter, better to cite Gagliardini2013

Thank you for pointing this out. We modified this sentence into "Gong et al (2017) adopted the four-step spin-up scheme (Gladstone et al., 2014) in inverse modelling using Elmer/Ice (Gagliardini et al., 2013), without testing the effect of initial temperature assumption on the inversion results."

l219: There is a step here that is not common, surface relaxation with C at its initial chosen value. What is done usually is the inversion, then the relaxation over about 15 years. I wonder the effect of the surface relaxation using a C that is far from reality...

"For cycle 1, the surface relaxation and first inversion are implemented with an initial temperature assumption (described below) and uniform basal friction coefficient of $10^{-4}$ MPa m$^{-1}$ a (following Gillet-Chaulet et al. (2012))." We clarified this in the text (Line 247-249).

Then we added another two cycles starting with surface relaxation from the initial geometry and simulated C from the previous cycle. Besides, the surface relaxation in each cycle was run for 0.2 yr, which is mentioned in Sect. 3.3. We also added a sentence in Sect. 3.3 (Line 200-202) "This is long enough to remove the non-physical spikes, but too short to significantly modify the geometry of the fast flowing regions of the Fleming Glacier"

l220: Basal sliding

As said above, we now use the consistent term "basal friction coefficient" in the whole text.

l225: Means you don't account for the modification of surface with relaxation at the beginning of the last two cycles ?

This seems to be a misunderstanding. Relaxation is carried out for each cycle, as stated. We point out that the relaxation of each cycle starts from the initial geometry. For each cycle, the modification of surface after relaxation (<25 m) is smaller than the uncertainty of the ice thickness based on the RMSE of difference between relaxed and observe surface elevations (see Table S1 in the supplementary material), which has been clarified in the Sect. 4.1 (Line 333-337). We feel this is quite clearly set out as it stands. This appears the sensible procedure to minimize the influence of any initial guess for $C$ in the first cycle on the relaxation, as raised by the reviewer above.

l228: Basal friction

Modified

l229: To your inverse method, not all of them

"inverse methods" -> "our inverse method"

l243: Don't say linear but rather Control

"linear" here is used to describe the way to generate the initial temperature field. The CONTROL experiment also contains a specific bedrock geometry (bed_bm). For clarity, we have rewritten as (Line 277): "the linear initial temperature distribution described above."

l246 to l265: I don't really understand the relevancy of this scenario. To me you should rather study the influence of the position of your ice front, since this is what you are not sure about with your hypothesis assuming ice front = grounding line.

The question is not as simple as ice front position, because division between intact ice shelf and iceberg/sea ice mélange is not clearly defined. Both ice front position and ice front pressure condition are relevant. The scenario here to adjust the external forcing on the calving front considers the uncertainties of ice thickness, bedrock depth, and backstress due to the ice mélange. But following the reviewer's suggestion, we have now added another sensitivity test to different ice front positions in Sect. 3.6 and Sect. 4.4. Note that we do not attempt to define a floating portion of the glacier.

l267: Results and discussion

Modified

l270: what do you call robustness here ? Replace drag by sliding. Rephrase please.

As we responded above to the comments regarding l56, "robustness" here means self-consistency. We think it is OK to use "robustness" here. To clarify it, we changed the sentence into "The evaluation criteria are the robustness of simulated basal friction coefficient distribution to experiment design and the mismatch between the simulated and observed surface velocities."

"drag"->"friction"

l273: There are only 3 TEMP experiments, be more clear

Modified. "the four TEMP experiments " -> "the CONTROL experiment and three TEMP experiments "

l275: Here what we need to have is a metric like the RMS, otherwise this is only a maximum value that is not representative of the rest of the data.

Thanks for the reviewer's suggestion. We have added this in Table S1 in the supplementary material. We calculated the root mean square difference (RMSD) of the difference between the relaxed and observed free surface for the fast flowing regions (>1500 m/yr). The RMSDs in elevation of all the experiments are all < 25 m.

l277: I don't understand what you say here ?

As mentioned in Sect. 3.3, the surface relaxation was used to remove the non-physical spikes in the initial observed surface DEM, caused for example by observational uncertainties of the surface or bedrock data and/or by the resolution discrepancy between mesh and geometry data. However, the surface relaxation cannot avoid systematic coherent changes in the surface near the ice front. To discuss the sensitivity of inverse modeling to this systematic change, we adopted different ice front boundary conditions in Sect. 4.4, which led to different changes in glacier surface during the surface relaxation. We modified this sentence (Line 336-337) "However, the systematic changes generated at the ice front during the surface relaxation may have effect on the inverse modeling, and this is further discussed in Sect. 4.4."

l279: This is quite difficult to evaluate the differences between the different experiments in your maps. I would recommend to the relative differences with a reference experiment.

Thanks for your suggestion. We plotted the relative differences between TEMP1-3 and CONTROL in Fig. S4, but the differences were mainly dominated by the slow-flowing areas. So we computed the RMSDs for $C$ (Table S2) and magnitudes of simulated basal velocity (Table S3) between TEMP1-3 and CONTROL for the fast flowing regions ($> 1500$ m yr$^{-1}$) in each cycle to evaluate the consistency of these experiments. The RMSD of magnitude of observed and simulated surface velocity for each experiment is also computed (Table S5).

l281: Figures should be ordered differently, such vertically Control, Temp1, Temp2, Temp3, this is otherwise very difficult to follow.

We think it is alright to put the figures horizontally as long as we keep the consistency of all figures in this text. Forcing more than three columns into the plots will make them smaller and harder to distinguish features properly. We changed the vertical ordering of different experiments and put CONTROL at the first row for each figure as requested even though it can make the trends in sensitivity more difficult to discern.

l283: Looking at Schaffer2012, it does not seem to me that the dependence of their model to temperature scenario is smaller than yours... You do need to quantify your differences, because this is really not clear.

See comments to l279.

l286: They showed a non influence onto the modelled surface velocity, not the friction coefficient, or I misread their paper... Their Fig8 shows the differences in terms of basal friction coefficient, but this slightly affect surface velocity as the inverse model tends to minimize the differences.

In Sect. 4.6 of Schäfer et al. (2012), they showed that the temperature scenario did not affect both surface and basal simulated velocities. So they made the conclusion that the obtained basal drag coefficients in their case did not depend strongly on the temperature.

l289 to l291: Already said, please avoid repetitions. You are in the result and discussion, thus adding other unnecessary stuff is only distracting the reader.

The reviewer seems to have lost track of which parts of the figure we are discussing. Having discussed the differences between results after a first cycle, we are moving to discuss the extent to which an additional cycle (and in due course a third cycle) reduces the dependence on the assumed initial englacial temperature distribution. We think it is necessary and appropriate to explain here why we implement the further cycle. We could understand that the remarks about Vestfonna modeling seem being a little repetitive and we shortened them.

l283: I think this is normal to have different results if you choose a sort of outlier in your initial state, like -20 degrees everywhere for the initial state. I don't think you discussed this as a comparison with the final result? Is -20 in the range of this final result?

Thanks for the suggestion. We agree that in a single cycle it is normal to expect "outliers", that is to say a lack of robustness between the results. So we computed the RMSD of the difference between the simulated temperature and the initial

temperature assumption for each cycle (Table S4). It shows that the experiment TEMP1 (beginning with -20 C) still shows notable differences to other simulations, even after three cycles. "Given this choice of preferred temperature initialization (CONTROL), and the significant difference between this and the cold initialization (TEMP1), we argue that TEMP1 likely deviates furthest from an ideal temperature initialization, and that such a large initial deviation would require more than three cycles to converge on a basal friction coefficient distribution. " This sentence has been added in the main text in Line 366-370.

l291: Drag - > friction coefficient

Modified.

l295: Could you quantify your sticky spots ?

Yes, we have modified this sentence into "However, for experiments CONTROL and TEMP2, the isolated sticky points ~3-5 km upstream of the ice front (with horizontal scale around ~1 km and peak basal friction coefficient of around $6\times10^{-5}$ MPa m$^{-1}$ yr) mostly decrease or disappear from the first cycle (Figs. 5a, 5g) to the second cycle (Figs. 5b, 5h)" (Line 350-353).

l296: "therefore..." remove this as this was already written

This is actually a new point. Here we try to explain the motivation of running the third cycle. To clarify this, we modify this sentence: "Therefore, a third cycle was implemented to test whether a two-cycle spin-up scheme was enough to reduce the dependence on the initial temperature assumptions." (353-355).

l300: You should say Control instead of linear scenario

Here we are not talking about the CONTROL simulations rather the scenario with linear initial temperature.

l306 to l308: Third time I see this in the paper, remove repetitions please.

We have deleted the earlier occurrence of a similar sentence in response to comment l289-l291. But this is the appropriate place for the Vestfonna discussion.

l306: The low impact is on modelled surface velocity. There is an impact on basal friction coefficient (or basal drag as they say)

They said the low impact on both the modeled surface and basal velocity, and the basal drag coefficients does not strongly depend on the temperature (Sect. 4.6, Sect. 5, and Fig. 13 in Schäfer et al. (2012)). So we are not wrong here.

l313: No need to say "inside the yellow contour" in the text

We think it is helpful to guide the reader to a specific aspect of the figure without referring to the figure caption. If it is strongly against the Cryosphere's style we could remove the remark.

l318: "shows that internal deformation": you should vertical deformation here.

Modified to "vertical shear deformation" to avoid confusion with strain thinning.

l319: I don't agree with this assertion. Vb=Vs in the fastest flowing areas. In between those you have an area with Vb much lower than Vs, but this matches the places where driving stress is much higher. So this is the driven stress that may drive the vertical deformation, not only the ice internal temperature... You need to rephrase.

This comment does not contradict our statement. The reviewer points out that the high vertical shear rate in our domain is a product of both high driving stress, and deformable (i.e. warm) ice. This is clearly true. We state that the basal state is sensitive to ice temperature – we have made no statement about the relevance or not of driving stress. To make it clearer, we modified the text (Line 390) to emphasize that we are referring to the region of high slope between the upstream and downstream basins, where the driving stress is high. Actually, these are regions of local higher basal shear stress than the surrounding regions, which is more directly relevant to shear deformation near the bed.

l330 to l332: not necessary because already mentioned

Moved to Sect. 4.1.

l334: Remove mentions to colors and rather explain with the physical parameters

See our response to comment l313. We modified this sentence into "in the fast-flowing region (>1500 m yr$^{-1}$, cyan contour in Fig. 7). The pattern in the region between the 1000 and 1500 m yr$^{-1}$ contours" (Line 422-423).

l340: I don't understand

We modified this sentence into "However, all three cases feature a low basal friction coefficient in the fast flow region (>1500 m yr$^{-1}$ in Fig. 7), which is approximately coincident with the FG downstream basin." (Line 425)

l345 to 347: Why mentioning the MISI in a paper that only deals with inversions, there is no point to me.

We agree that MISI cannot directly explain over- or under- estimation of velocities in an inversion. We deleted this sentence.

l347: basal friction

Modified.

l350: What is behind "it" ? The link with previous sentences is not quite clear. Rephrase please.

We deleted the sentence starting with "it means" and added one sentence before it. "One possible cause of the different basal friction coefficient distributions in these inversions might be the changed surface topography during the surface relaxation, especially near the ice front (Figs. S6)." (Line 431-433)

l355: Ok great, you calculated RMSEs. However, 1) you should have done it before (see previous comment above) and 2) please give us numbers.

Thanks for your suggestion. We have added the RMSD of each experiment in the text and Table S5 in the supplementary material.

l357: I guess this is justified by your RMSE. I think you should discuss more this result, as it suggest that using data taken over a short time range improve the results compared to Bedmap2, which is taken over larger time scales than a year... If I am not wrong.

Thanks for your point. We added few sentences here to clarify the reason why we chose BEDZC (Line 438-453)

l368 to l371: remove this as already written in the methods section

In the Sec. 3.6, we only discussed the reason for setting different ice front boundary conditions. Here we are talking about the possible reasons for the high friction spots near the front. So we don't think this comment should be removed. This is not an exact copy of the earlier section, and it gives context for the current discussion, given the emergence of the high friction spots in the simulations of the previous sections.

l371: You did not really have investigated the sensitivity to uncertainty to me. You only have tested two datasets, one being more accurate than the other by the way. The Mass conservation based inversion for bedrock is quite an efficient method to infer the bedrock (see Morlighem2014 NG)

Modified "bedrock uncertainty" into "bedrock datasets".

Here we are presenting a sensitivity study – we are not aiming to explore the full range of uncertainty. We have chosen different bedrock datasets that can be justified, and we carry out a sensitivity experiment using these datasets. It is true that this does not quantify the full range of possible outcomes as a response to bedrock uncertainty, but we are not claiming to do that.

It is also not true to say that in general the mass conservation method is "more accurate" than interpolation of direct observations. It may often be preferable, but there are many factors.

l382: This is the kind of things you need to check really. You may have the answer in the paper by Mouginot 2012 in the Journal Remote Sensing. It seems to be a combination between 2007 to 2009 data.

The epoch we quote in the paper (Line 131) was taken from the published information about the various contributions to MEaSUREs velocity datasets we used. The velocity data for the Fleming system is derived from the PALSAR (see the supplementary information in Rignot et al. (2011b)). The PALSAR measurements used in that paper covers coastal sectors north of 77.5° S in "Fall 2007 and 2008". We did check the paper you mentioned, but it did not give us extra information.

We have modified this sentence into "Regarding velocities, Friedl et al. (2018) presented evidence that an acceleration phase occurred between Jan-Apr 2008, but the surface velocity data used in this study was extracted from measurements in Fall 2007 and 2008 (Rignot et al., 2011b)." (Line 468-470).

l387: I really question the relevancy of this experiment. Why doing so as it seems to me that more relevant experiment would be to adjust the ice front position, where you have your uncertainty, and check the sensitivity of inversion results. This latter experiment would not change much the results to me, because over 1.5 km of ice shelf, you don't have much buttressing, but it would be more relevant than what you propose to me.

We have added the experiments of adjusting the ice front position in the Sect. 4.4 to partly address this comment.

l421: I don't understand, in what context ?

Here we mean "The lowered surface at the ice front in experiments IFBC1 and CONTROL is apparently the consequence of rapid deformation due to its own weight (longitudinal extension with locally high vertical shear) of an ice cliff, which is over 100 m higher than the control sea level" (Line 526-529).

l429: This is still about this experiment. To test such an amplitude in the influence of

sea levels in inversion results, you need to cite literature about what buttressing could be added from ice mélange (see Krug2014 by the way)...

We guess you mean Krug et al. (2015). This sentence is about presences/absence of the ice front high basal friction being connected with the ice front boundary conditions and not about the local driving stress modification in the relaxation step. We are trying here to address the effect of uncertainty in bed elevation rather than buttressing and mélange. We emphasized that the experiments with different sea levels represent some small uncertainty in the actual sea level, but is also a proxy for pressure variations due to thickness and bed uncertainty and mélange back stress (Line 302-304).

We calculated that ice mélange back force ($\sim$1.1e7 N m$^{-1}$) used to prevent the rotation of iceberg at the calving front (Krug et al., 2015) could account for the equivalent of up to $\sim$2.3 m sea level in terms of ice front boundary condition. We added this sentence in Line 513-514.

Figure 4: add relaxation time here

Added.

Figure 5 caption: Temp4 doesn't exist

Modified.

Figures in general: All the differences that you comment are not always visible. These are to me really tight differences so if you want to argue on this to underline the improvement that are brought by your 4 cycle spin up scheme, you should care more about the figures. Use relative differences between the Control and the other experiments.

Thanks for your suggestions. We hope it is understood that our study concerns the iteration of the original four step spin-up scheme of Gladstone et al (2014). We plotted the relative differences between TEMP1-3 and CONTROL in Fig. S4. We also computed the RMSDs of $C$ (Table S2) and of the magnitudes of simulated basal velocity (Table S3) between TEMP1-3 and CONTROL for the fast flowing regions (> 1500 m yr$^{-1}$) in each cycle to evaluate the consistency of these experiments. The RMSDs of magnitudes of observed and simulated surface velocity for each experiment is also computed (Table S5). We modified our analysis about the temperature simulations in Sect. 4.1 (Line 356-374).

Figures in general: please, for the readability order vertically your subplots like: Control, temp1, temp2, temp3

As we comment on l281, we changed the order of different experiments and put CONTROL at the first row for each figure.

Figure 7: Here is certainly a way to remove those zigzags discontinuity, I know Paraview is not user friendly for some stuff, but I don't think this is acceptable for a peer reviewed paper.

This figure has been moved into Fig. S4 in the supplementary material. We do not think the zigzag artefacts interfere with the interpretation of the figure, but can try to improve it if the editor regards it as important

References

[revised manuscript text omitted]
 adjust the sea level by 10 m from hydraulic equilibrium to test the sensitivity of the inverse modeling to the ice front boundary condition. Firstly, we assume an ocean pressure at the ice front computed using the sea level mentioned in Sec. 2.1. We further simulate two alternative scenarios for the sea level used in the simulations to calculate ocean pressure: IFBC1 with a sea level of 5 m and IFBC2 with a sea level of 25 m. Another extreme scenario (IFBC3, Table 2) is adopted here by setting the ice front pressure to the ice overburden:

$$P_i(z) = \rho_i g(z_s - z)$$
(57)

where $P_i(z)$ is the pressure at the ice front as a function of height $z$, $\rho_i$ is ice density (Table 1), $g$ is the gravitational constant (Table 1), and $z_s$ is the height of ice upper surface at the ice front. This is the pressure that would be imposed by a hypothetical undeforming continuation of the advancingneighboring glacier, and imposes zero normal strain rate at the ice front. The ice surface elevation $z_s$ at the front is ~115 m, approximately 100 m above actual sea level. The total vertically integrated pressure imposed by this condition is equivalent to a sea level of ~60 m, although the vertical distribution of pressure is different todiffers from an ocean pressure condition. Experiments IFBC1, IFBC2 and IFBC3 differ from CONTROL only in their ice front boundary condition.

In our model domain we assume the 2008 grounding line is consistent with the 1996 grounding line, which has an error of several km on fast-moving ice (Rignot et al., 2011a) and might have changed since 1996. The frontal surface elevation is from the SPOT DEM data in Jan 2008, which shows the ice front position is ~1.5 km downstream of the 1996 grounding line position. Since such a narrow residual ice shelf was considered unlikely to have a major influence we constructed the model geometry to have the ice front coincide with the 1996 grounding line for simplicity, i.e. all ice is considered grounded.

**4 Results and discussions**

The main focus of the current study is the sensitivity of the inversion to the variations of three five factors: temperature initialization, enhancement factor, bed topography, ice front positions, and ice front stress balanceoceanic pressure boundary condition. The evaluation

criteria are the robustness of simulated basal friction coefficient distribution to experiment design and the mismatch between the simulated and observed surface velocities.

**4.1 Sensitivity to initial temperature**

We present the results for the inferred basal friction coefficients from the CONTROL and  three TEMP experiments (Sect. 3.6, Table 2) for the WIS-FG system in Fig. 5. The 2008 ice velocity contours are added as visual references for comparing the basal friction coefficient patterns in the regions of fast flow, since the largest observed ice velocity changes occurred in fast flowing outlet  regions (Mouginot et al., 2014; Walker and Gardner, 2017).

In each cycle, the root-mean-square deviation (RMSD, sometimes also called root-mean-square error) between the relaxed and the observed surface was < 25 m (see Table S1 in the supplementary material), smaller than the ice thickness uncertainty (> 50 m) used in this study. However, the systematic changes generated at the ice front during the surface relaxation may have an effect on the inversion, and this is further discussed in Sect. 4.4.

 After the first cycle (left column, Fig. 5), results showed different patterns of basal friction coefficient for each experiment, especially in the fast-flowing regions with surface velocity exceeding 1000 m yr⁻¹ (yellow contour in Fig. 5). The basal friction coefficients from TEMP2 (Fig. 5g) and CONTROL (Fig. 5a) share  similar sticky spots around the ice front, and some isolated sticky spots ~3-5 km upstream of the ice front, but  TEMP1 (Fig. 5d) and TEMP3 (Fig. 5j) display different patterns, indicating dependence on the initial temperature assumption. The RMSDs of key properties are computed to evaluate the consistency of these experiments (Table S2-S5).~~This is in contrast to a similar inverse study on the Vestfonna ice cap (Schäfer et al., 2012), which showed little impact of temperature distribution on the basal sliding coefficient. That was due to a low contribution of ice deformation to ice motion compared to the basal sliding (Schäfer et al., 2012). We return to this contrast after considering the effect of the second and third cycles of our spin-up.~~

[revised manuscript text omitted]

~~The deeper retrograde bed in the CONTROL simulation may indicate increased vulnerability to marine ice sheet instability, and more overestimation of surface velocity is found around the grounding line (Fig. 8d). One possible cause of the different basal shear stress in these inversions might be the increased slope caused by the surface relaxation. However, we find the inversion process is not sensitive to the surface relaxation, and this is further discussed in Sect. 4.3. It means a high accuracy bedrock topography data is very important for inverse modeling owing to the fact that the bedrock resolution around the grounding line determines the ice dynamics .ssesclearbestlthisWe compute troot mean square errors (E), and find the RMSE ofismarginally larger than the RMSE ofWhile bmechanism~~approach (which is independent of surface geometry) and the bed geometries are accordingly more similar to each other than they are to CONTROL (see Fig. 2 b-d). However, BEDZC maintains better internal consistency with the

2008 surface elevation, since it results in the mass conserving ice thickness $H_{mc}$ being employed, whereas, by the construction of bed_mc (Eq. 2), the ice thickness in BEDMC is not entirely consistent with mass conservation, although still a more physically motivated interpolation than bed_bm in CONTROL. The BEDMC and BEDZC ice thicknesses clearly differ by the difference between the Bedmap2 and 2008 DEMs, which should be greatest in areas of greatest lowering, and as we see BEDMC provides a useful sensitivity test case. Since bed_zc is extracted from the accurate and contemporary DEM2008, it should also incorporate into the bed geometry (via $H_{mc}$) more detail from the then current surface, compared to bed_mc, extracted from Bedmap2's surface DEM, which was generated over a longer time range data used in current study than BEDMC.. Therefore, bed_zc is suggested as the best current bedrock elevation data for further ice sheet modelling of the WIS FGLFG system.

**4.3 4 Sensitivity to ice front position and boundary condition**

All the inversions presented so far feature both a bandsome sticky spots of with high basal dragfriction coefficient near the ice front of the Fleming Glacier (right column of Fig. 5 and left column of Fig. 87) and a similar localized overestimate of upper surface velocities at the ice front (right column of Fig. 6 and middle column of Fig. 8). We now consider causes for possible uncertainties about in the force applied to the ice front, and whether the high basal friction near the ice front is likely to be a feature of the real system or emerges from the inversion process as a compensating response to incorrect boundary forcing by the inversion process. These possible causes include uncertainty in local bedrock elevation (or equivalently ice thickness), uncertainty in observed sea level, uncertainty in exact ice front position and grounding line position, uncertainty in surface velocity, and uncertainty in potential backstress due to ice mélange and/-or grounded icebergs in contact with the ice front. The sensitivity to various bedrock uncertainty datasets has been discussed in Sec. 4.23. In our model domain we assume the 2008 grounding line is consistent with the 1996 grounding line, which has an error of several km on fast-moving ice (Rignot et al., 2011a) and might have changed since 1996. The frontal surface elevation is from the SPOT DEM data in Jan 2008, which shows the ice front position is ~1.5 km downstream of the 1996 grounding line position. Since such a narrow residual ice shelf was considered unlikely to have a major influence we constructed the model geometry to have the ice front coincide with the 1996 grounding line for simplicity, i.e. all ice is considered grounded.In this frameworkBy assuming the ice front position to coincide with the 1996 grounding line, uncertainty about the bedrock depth at the ice front feeds in-to significant uncertainty in the total restraining force from ocean pressure. Regarding velocities, Friedl et al. (2018) presented evidence that an acceleration phase occurred on the Fleming Glacier around between MarchJan-Apriil 2008, but we are not sure the specific month of the surface velocity data used in this the current study was extracted from measurements in Fall 2007 and 2008 (Rignot et al., 2011b). ThisIt means the surface velocity data, which is provide the target to be matched by the control inversione process, might not be consistent with the DEM data used here (acquired in Jan 2008).

To explore the influence of these different sources of uncertainty, we adopt different ice front positions and effective sea level heights within our vertical reference frame to apply a range of ocean pressures to the ice front as described in Sect. 3.6 (IFBC1-3 and IFP1-2, Table 2).

Experiments with different ice front positions (IFP1-2 in Table 2) directly affect the ice thickness and bed elevation at the ice front, which affects the ice front pressure condition. The simulated basal friction coefficients (left column in Fig. 8) show that the high sticky spots near the ice front migrate with the ice front position but with different patterns. The experiment IFP1 with a seaward shifted ice front position shows a decrease in magnitude of the high friction spots (Fig. 8b) and a better match with the observed velocity (Fig. 8e), while the IFP2 with a retreated ice front shows an increased $C$ (Fig. 8c) and worse surface velocity match (Fig. 8f) compared with CONTROL experiment (Figs. 8a, 8d). In experiment IFP1,

thinner ice at the ice front leads to a relatively smaller ice velocity compared with CONTROL, so the model does not need to increase *C* to match the observed surface velocity. This does not mean that ice front position in IFP1 is more accurate than CONTROL, since the time inconsistency of surface DEM data, ice front and grounding line position, and surface velocity data is the obstacle to obtaining a reliable basal friction pattern. Therefore, we speculate that some of the high basal friction spots near the ice front are artefacts. However, we do not exclude the possibility of high basal friction spots caused by the pinning points located at the 1996 grounding line, which is also proposed by Friedl et al. (2018). An accurate location of the ice front and grounding line is clearly important for inverse modelling of fast flowing glaciers like the Fleming Glacier.

A higher sea level in the ice front boundary condition imposes a higher pressure at the ice front, i.e. a higher total retarding force, and we impose these different boundary conditions as a proxy for the sources of uncertainty discussed above.

Basal friction coefficients *C* simulated from the IFBC1-2 and CONTROL experiments (Figs. 9a-c) present  similar patterns but differ systematically around the ice front regions  (within ~1 km of the grounding line). Experiments with higher sea levels display smaller *C* there (Fig. 9, left column) and provide a better match between modeled and simulated surface velocities (Fig. 9, middle column), which is consistent with the computed RMSD of the surface velocity mismatch (Table S5). If the applied ice front boundary condition underestimates the real world forcing, the inversion process will compensate by increasing the basal friction in this region. ~~However, the large vertical shear strain rate imposes a limit to how much increasing basal drag can reduce the surface velocity, which could explain why the mismatch between the modeled and observed velocity is still large in the narrow band near the ice front (Fig. 9, middle column). For the fast flowing region (velocity > 1500 m a⁻¹), the decreased basal shear force from IFBC2 to IFBC3 (~1.1×10¹¹ N) roughly matches the increased the ice front pressure over a 6 km length of ice front (~2.8×10¹¹ N).~~

[revised manuscript text omitted]

---

## Author Response (AR2)

Editor's comments

We thank the Editor Ben Smith for the positive and constructive suggestions to improve our paper. We have addressed all comments below. The line numbers in the responses are based on the revised manuscript without track changes.

Comments to the Author:

Editor's note on tc-2017-241

Thanks to the authors for their revisions to this paper in response to the referees' comments. Looking over the comments, I feel that the manuscript would benefit from restructuring to address referee 2's comments that the text tends to repeat itself, and that the figures show too many versions of very similar model fields. The authors' responses have defended the repetition, but I recommend reducing the number of panels in figures 5 and 9 (see below) and moving figure 7 to the supplement. The corresponding descriptions in the text should also be reduced and/or moved to the supplement.

Thanks for the editor's suggestions. We have modified Fig. 9 as the editor suggested (see below). Fig. 7 has been moved to the supplementary material as Fig. S6. The relevant text has been modified. As discussed below we would prefer to leave Fig. 5 and the related discussion as it stands.

Both reviewers commented on the choice of a non-zero sea level. The authors need at least a sentence to explain their choice of WGS84, rather than a geoid, as their reference surface, and to evaluate the magnitude of the problems this might cause. This is a potential problem in calculating the driving stress (which should be determined by the surface gradient with respect to the geoid) and in calculating heights above flotation (as is relevant to the companion paper). Given that the authors are using WGS-84 as their reference surface, it is not obvious that the height of the sea surface in BEDMAP-2 is the appropriate reference surface, so the authors should refer to the BEDMAP-2 documentation to see how what the ocean surface there is intended to represent: is it a mean sea-surface height or the height at some tidal phase (i.e. mean low-low tide)?

The topographic datasets in this study originate in different (geoid) reference frames. We chose WGS84 ellipsoid heights (for which convenient conversion data were available) rather than a geoid to make sure all the elevation datasets used in this study are in the same reference system. The SPOT DEM and ASTER DEM used the EGM96 geoid (Lemoine et al. 1998) as the height reference, while BEDMAP-2 used the GL04C geoid (Förste et al. 2008) as the absolute height reference. To clarify this, we added a sentence at Line 88-91 "Both SPOT and ASTER DEM products used the EGM96 geoid (Lemoine et al. 1998) as the height reference. However, the bed elevation data from Bedmap2 dataset (Fretwell et al. 2013) adopted the EIGEN-GL04C geoid (Förste et al. 2008) as its height reference, and we chose to convert all

the elevation datasets to the WGS84 ellipsoid. " We also modified the sentence at Line 308-310, "In addition to the ice front position, there are other sources of uncertainty in the vicinity of the ice front: ice thickness, bedrock depth, height conversion from geoid to ellipsoid, and backstress due to the presence of ice mélange.", and the sentence at Line 477-481, "These possible causes include uncertainty in local bedrock elevation (or equivalently ice thickness), uncertainty in the geoid-ellipsoid height conversion, uncertainty in observed sea level, uncertainty in exact ice front position and grounding line position, uncertainty in surface velocity, and uncertainty in potential backstress due to ice mélange and/or grounded icebergs in contact with the ice front.".

We converted all the BEDMAP-2 datasets used in this study from the GL04C geoid to the WGS84 ellipsoid based on the conversion file (gl04c_geoid_to_WGS84.tif) provided by Fretwell et al. (2013). We have clarified this at Line 108-110, "The first is from the Bedmap2 dataset (Fretwell et al. 2013) with a resolution of 1 km (hereafter bed_bm; Fig. 2b), which we converted from the EIGEN-GL04C geoid (Förste et al. 2008) to WGS84 ellipsoid heights. "

We have checked that both geoid-ellipsoid separation fields vary very slowly spatially compared to the surface elevation of the ice sheet, so that we do not expect any significant change in the computed surface slope that enters the driving stress calculations from mapping the geoid-based elevations into the ellipsoidal frame. Ice thickness is preserved in converting the datasets to the ellipsoid reference frame. One additional issue in using ellipsoidal heights rather than elevation relative to the geoid is that sea surface height is required for the water pressure boundary condition at the ice front, rather than being implicitly zero. This is already mentioned in Sect 3.6 in the description of the CONTROL experiment where we mention using the observed sea level height (Line 310-320) while the specification of other sea level values forms part of the sensitivity test to ice front boundary conditions (see Sect. 4.4). The sea level is obviously relevant to the heights above flotation in the companion paper – and the relevant equations there explicitly include the sea level. To clarify these matters, we added one sentence at Line 96-102, "Both geoid-ellipsoid separation fields vary very slowly spatially compared to the surface elevation of the ice sheet, so that we do not expect any significant change in the computed surface slope that enters the driving stress calculations from mapping the geoid-based elevations into the ellipsoidal frame. Ice thickness is preserved in converting the datasets to the ellipsoid reference frame (see Sect. 2.2). Clearly, the sea level height in the ellipsoidal reference frame enters the calculation of ocean water pressure on the ice front explicitly, as we discuss under experimental design in Sect 3.6 and Sect. 4.4.".

Also, while we found no relevant information characterising the sea surface height in the BEDMAP2 documentation, we remind the Editor that the basis of the BEDMAP2 reference frame is the EIGEN-GL0C4 geoid (Förste et al. 2008). As one would expect from the standard definition of the geoid, construction of EIGEN-GL0C4 involves mean sea surface height - to quote their Conclusions:

*A new satellite-only gravity field model, EIGEN-GL04S1, complete up to degree and order 150 has been inferred from GRACE and LAGEOS data. Using this satellite-only model as a starting point, a new combined global gravity fieldmodel EIGEN-GL04C, complete up to degree and order 360, has been developed, incorporating surface gravity data including newly available or improved data sets of the Arctic, Antarctica and North-America and improved mean sea surface heights from altimetry processing at GFZ.*

At least one referee commented on the linear sliding law. The authors should make clear in the text that the sliding law is only used as a numerically convenient tool for calculating the basal shear stress.

Thanks for the editor's suggestion. Reviewer 1 said:

Line 163: What is your justification for using a linear sliding law?

We replied: Different sliding laws in inverse modeling will not change the inversed basal shear stress distribution, and it will just lead to different basal friction coefficients based on different sliding law. In diagnostic studies that invert to find the basal shear stress which gives the best agreement with observed surface velocities, the choice of sliding "law" is not relevant provided that the required stress can be generated by adjustments of the parameters in the sliding law – in this case the coefficient C. The inversion procedure modifies C to modify stress – adjusting the momentum balance. That solution of the Stokes equation provides an updated estimate of basal velocity – which enters the next cycle of the inversion search. The question does remain whether this is physically suitable relationship to apply when the system is evolving, but this is not relevant here. So we adopted the simplest sliding law here following Gillet-Chaulet et al. (2012), Gagliardini et al. (2013). We clarified this in the text.

To clarify this, we altered one sentence at Line 202-208, "Considering that in this diagnostic study the sliding law is only used as a numerically convenient tool for calculating the basal shear stress, a simple linear sliding law following Gillet-Chaulet et al. (2012), Gagliardini et al. (2013) is applied on the bottom surface:

$$\tau_b = Cu_b \qquad\qquad\qquad\qquad\qquad\qquad\qquad\qquad (5)$$

where C, the basal friction coefficient, is used as the adjustable parameter in the inversion scheme described below."

We would be happy to provide more detail in the main text if the editor feels this is needed.

I recommend that the authors convert figure 5 to include only two versions of the temperature initialization (the one selected, and a second version, of their choice, that demonstrates unacceptable behavior), and only the first and last steps of the temperature iteration. The intermediate steps and the full range of models tested can be described, very briefly, in the text.

We can certainly comply with this recommendation if the editor insists, but we feel it undermines the presentation of the multiple cycle spin-up procedure. We think demonstrating the extent to which multi-cycle spin-up reduces dependence on assumed initial temperatures will be of broad interest to modellers, beyond our other location-specific sensitivity studies. If we must modify Fig. 5 as the editor suggests, we would want to include the original figure and relevant discussion in the supplementary material but we would prefer to have it more directly citable in the main paper. We could show the most disparate cases - simulated basal friction coefficient distributions for CONTROL and TEMP1 (T = -20 C) - in the main text, and the results of other two experiments TEMP2 and TEMP3 in the supplementary material but this fragmentation of the examples seems clumsy and the loss of discussion of the progression through the cycles regrettable.

I recommend that the authors remove the right-hand column of figure 9 (the relaxation height difference). This column is given only cursory attention in the text.

Thanks for the editor's suggestion. We have removed the right column of Fig. 8 (the original Fig. 9) and modified the relevant text.

For all figures (5-9) with multiple panels, the rows and the columns of the figures should be labeled, so that the reader does not need to refer to the letter key in the captions to interpret each panel.

Thanks for the editor's suggestion. We have labeled the rows and columns of those figures (Figs. 5-8, and figures in the supplementary).

These changes are somewhat substantial, and I would like to look over the authors' response to them before I consider whether it needs a re-review or not.

Best

Ben Smith

At the ice front, the normal component of the stress where the ice is below sea level is equal to the hydrostatic water pressure exerted by the ocean. The uncertainties of ice thickness and bedrock topography, the low accuracy of ice front and grounding line locations, and the possible buttressing on the ice front by partly detached icebergs and ice mélange (see Fig. 1c) would affect the calculation of ocean forcing there. Accordingly, wWe 
[revised manuscript text omitted]

~~The deeper retrograde bed in the CONTROL simulation may indicate increased vulnerability to marine ice sheet instability, and more overestimation of surface velocity is found around the grounding line (Fig. 8d). One possible cause of the different basal shear stress in these inversions might be the increased slope caused by the surface relaxation. However, we find the inversion process is not sensitive to the surface relaxation, and this is further discussed in Sect. 4.3. It means a high accuracy bedrock topography data is very important for inverse modeling owing to the fact that the bedrock resolution around the grounding line determines~~

[revised manuscript text omitted]

---

## Author Response (AR3)

Editor's comments

We thank the Editor Ben Smith for the positive and constructive suggestions to improve our paper. We have addressed the comments below.

Comments to the Author:

Editor's note on tc-2017-241

Thanks for your responses to my comments on the second version, and please excuse my inattention in missing your response to the referee about the nature of the basal-friction coefficient. I see that you answered that question well in your response.

Please consider moving the third column of table S5 to table 2 in the main text of the paper. The RMSE of the surface velocity seems like an important metric of the success or failure of each inversion scenario, and helps place information needed to interpret the results of the paper in the text rather than in the supplement. Once this is done, I think the paper should be ready to go.

Thanks for the editor's suggestions. We agree that the RMSE of the surface velocity plays an important role in interpreting the results of the paper in the text. We have moved the third column of table S5 to table 2 in the main text. The relevant text has been modified.

[revised manuscript text omitted]
 adjust the sea level by 10 m from hydraulic equilibrium to test the sensitivity of the inverse modeling to the ice front boundary condition. Firstly, we assume an ocean pressure at the ice front computed using the sea level mentioned in Sec. 2.1. We further simulate two alternative scenarios for the sea level used in the simulations to calculate ocean pressure: IFBC1 with a sea level of 5 m and IFBC2 with a sea level of 25 m. Another extreme scenario (IFBC3, Table 2) is adopted here by setting the ice front pressure to the ice overburden:

$$P_i(z) = \rho_i g(z_s - z)$$
(57)

where $P_i(z)$ is the pressure at the ice front as a function of height $z$, $\rho_i$ is ice density (Table 1), $g$ is the gravitational constant (Table 1), and $z_s$ is the height of ice upper surface at the ice front. This is the pressure that would be imposed by a hypothetical undeforming continuation of the advancingneighboring glacier, and imposes zero normal strain rate at the ice front. The ice surface elevation $z_s$ at the front is ~115 m, approximately 100 m above actual sea level. The total vertically integrated pressure imposed by this condition is equivalent to a sea level of ~60 m, although the vertical distribution of pressure is different todiffers from an ocean pressure condition. Experiments IFBC1, IFBC2 and IFBC3 differ from CONTROL only in their ice front boundary condition.

In our model domain we assume the 2008 grounding line is consistent with the 1996 grounding line, which has an error of several km on fast-moving ice (Rignot et al., 2011a) and

~~might have changed since 1996. The frontal surface elevation is from the SPOT DEM data in Jan 2008, which shows the ice front position is ~1.5 km downstream of the 1996 grounding line position. Since such a narrow residual ice shelf was considered unlikely to have a major influence we constructed the model geometry to have the ice front coincide with the 1996 grounding line for simplicity, i.e. all ice is considered grounded.~~

**4 Results and discussions**

The main focus of the current study is the sensitivity of the inversion to the variations of  five factors: temperature initialization, enhancement factor, bed topography, ice front positions, and ice front oceanic pressure boundary condition. The evaluation criteria are the robustness of simulated basal friction coefficient distribution to experiment design and the mismatch between the simulated and observed surface velocities.

**4.1 Sensitivity to initial temperature**

We present the results for the inferred basal friction coefficients from the CONTROL and  three TEMP experiments (Sect. 3.6, Table 2) for the WIS-FG system in Fig. 5. The 2008 ice velocity contours are added as visual references for comparing the basal friction coefficient patterns in the regions of fast flow, since the largest observed ice velocity changes occurred in fast flowing outlet regions (Mouginot et al., 2014; Walker and Gardner, 2017).

In each cycle, the root-mean-square deviation (RMSD, sometimes also called root-mean-square error) between the relaxed and the observed surface was <  25 m (see Table S1 in the supplementary material), smaller than the ice thickness uncertainty (> 50 m) used in this study. However, the systematic changes generated at the ice front during the surface relaxation may have an effect on the inversion, and this is further discussed in Sect. 4.4.

 After the first cycle (left column, Fig. 5), results showed different patterns of basal friction coefficient for each experiment, especially in the fast-flowing regions with surface velocity exceeding 1000 m yr$^{-1}$ (yellow contour in Fig. 5). The basal friction coefficients from TEMP2 (Fig. 5g) and CONTROL (Fig. 5a) share  similar sticky spots around the ice front, and some isolated sticky spots ~3-5 km upstream of the ice front, but  TEMP1 (Fig. 5d) and TEMP3 (Fig. 5j) display different patterns, indicating dependence on the initial temperature assumption. The RMSDs of key properties are computed to evaluate the consistency of these experiments (Tables 2, S2-S5).~~This is in contrast to a similar inverse study on the Vestfonna ice cap (Schäfer et al., 2012), which showed little impact of temperature distribution on the basal sliding coefficient. That was due to a low contribution of ice deformation to ice motion compared to the basal sliding (Schäfer et al., 2012). We return to this contrast after considering the effect of the second and third cycles of our spin up.~~

[revised manuscript text omitted]

~~The deeper retrograde bed in the CONTROL simulation may indicate increased vulnerability to marine ice sheet instability, and more overestimation of surface velocity is found around the grounding line (Fig. 8d). One possible cause of the different basal shear stress in these inversions might be the increased slope caused by the surface relaxation. However, we find the inversion process is not sensitive to the surface relaxation, and this is further discussed in Sect. 4.3. It means a high-accuracy bedrock topography data is very important for inverse modeling owing to the fact that the bedrock resolution around the grounding line determines~~

the ice dynamics . Comparisons of the distributions of velocity mismatch and of *C* between BEDZC and BEDMC does not provide a clear direct insight into which is the best more accurate basal geometry for modelling this the Fleming system. We compute tThe computed root mean square errors (RMSED) of the velocity mismatch for the regions with velocity >1500 m yr$^{-1}$ (Table 2) is only slightly higher for , and find the RMSE of BEDMC (62.60 m yr$^{-1}$) is than for marginally larger than the RMSE of BEDZC (61.78 m yr$^{-1}$), and both are much lower than CONTROL. Both BEDMC and BEDZC use the 2008 surface DEM and this improvement over the Bedmap2 surface DEM in CONTROL appears significant, even before turning to the matter of ice thickness. While bBoth cases use the ice thickness extracted using the mass conservation mechanismapproach (which is independent of surface geometry) and the bed geometries are accordingly more similar to each other than they are to CONTROL (see Fig. 2b-d). However, BEDZC maintains better internal consistency with the 2008 surface elevation, since it results in the mass conserving ice thickness H$_{mc}$ being employed, whereas, by the construction of bed_mc (Eq. 2), the ice thickness in BEDMC is not entirely consistent with mass conservation, although still a more physically motivated interpolation than bed_bm in CONTROL. The BEDMC and BEDZC ice thicknesses clearly differ by the difference between the Bedmap2 and 2008 DEMs, which should be greatest in areas of greatest lowering, and as we see BEDMC provides a useful sensitivity test case. Since bed_zc is extracted from the accurate and contemporary DEM2008, it should also incorporate into the bed geometry (via H$_{mc}$) more detail from the then current surface, compared to bed_mc, extracted from Bedmap2's surface DEM, which was generated over a longer time range data used in current study than BEDMC.. Therefore, bed_zc is suggested as the best current bedrock elevation data for further ice sheet modelling of the WIS-FGLFG system.

**4.3 4 Sensitivity to ice front position and boundary condition**

All the inversions presented so far feature both a bandsome sticky spots of with high basal dragfriction coefficient near the ice front of the Fleming Glacier (right column of Fig. 5 and left column of Fig. 8S6) and a similar localized overestimate of upper surface velocities at the ice front (right column of Fig. 6 and middle column of Fig. 8). We now consider causes for possible uncertainties about in the force applied to the ice front, and whether the high basal friction near the ice front is likely to be a feature of the real system or emerges from the inversion process as a compensating response to incorrect boundary forcing by the inversion process. These possible causes include uncertainty in local bedrock elevation (or equivalently ice thickness), uncertainty in the geoid-ellipsoid height conversion, uncertainty in observed sea level, uncertainty in exact ice front position and grounding line position, uncertainty in surface velocity, and uncertainty in potential backstress due to ice mélange and/-or grounded icebergs in contact with the ice front. The sensitivity to various bedrock uncertainty datasets has been discussed in Sec. 4.23. In our model domain we assume the 2008 grounding line is consistent with the 1996 grounding line, which has an error of several km on fast-moving ice (Rignot et al., 2011a) and might have changed since 1996. The frontal surface elevation is from the SPOT DEM data in Jan 2008, which shows the ice front position is ~1.5 km downstream of the 1996 grounding line position. Since such a narrow residual ice shelf was considered unlikely to have a major influence we constructed the model geometry to have the ice front coincide with the 1996 grounding line for simplicity, i.e. all ice is considered grounded.In this frameworkBy assuming the ice front position to coincide with the 1996 grounding line, uncertainty about the bedrock depth at the ice front feeds in-to significant uncertainty in the total restraining force from ocean pressure. Regarding velocities, Friedl et al. (2018) presented evidence that an acceleration phase occurred on the Fleming Glacier around between MarchJan-Apri l 2008, but we are not sure the specific month of the surface velocity data used in this the current study was extracted from measurements in Fall 2007 and 2008 (Rignot et al., 2011b). ThisIt means the surface velocity data, which is provide the target to be matched by the control inversione process, might not be consistent with the DEM data used here (acquired in Jan 2008).

To explore the influence of these different sources of uncertainty, we adopt different ice front positions and effective sea level heights within our vertical reference frame to apply a range of ocean pressures to the ice front as described in Sect. 3.6 (IFBC1-3 and IFP1-2, Table 2).

Experiments with different ice front positions (IFP1-2 in Table 2) directly affect the ice thickness and bed elevation at the ice front, which affects the ice front pressure condition. The simulated basal friction coefficients (left column in Fig. 7) show that the high sticky spots near the ice front migrate with the ice front position but with different patterns. The experiment IFP1 with a seaward shifted ice front position shows a decrease in magnitude of the high friction spots (Fig. 7b) and a better match with the observed velocity (Fig. 7e), while the IFP2 with a retreated ice front shows an increased $C$ (Fig. 7c) and worse surface velocity match (Fig. 7f) compared with CONTROL experiment (Figs. 7a, 7d). In experiment IFP1, thinner ice at the ice front leads to a relatively smaller ice velocity compared with CONTROL, so the model does not need to increase $C$ to match the observed surface velocity. This does not mean that ice front position in IFP1 is more accurate than CONTROL, since the time inconsistency of surface DEM data, ice front and grounding line position, and surface velocity data is the obstacle to obtaining a reliable basal friction pattern. Therefore, we speculate that some of the high basal friction spots near the ice front are artefacts. However, we do not exclude the possibility of high basal friction spots caused by the pinning points located at the 1996 grounding line, which is also proposed by Friedl et al. (2018). An accurate location of the ice front and grounding line is clearly important for inverse modelling of fast flowing glaciers like the Fleming Glacier.

A higher sea level in the ice front boundary condition imposes a higher pressure at the ice front, i.e. a higher total retarding force, and we impose these different boundary conditions as a proxy for the sources of uncertainty discussed above.

Basal dragfriction coefficients $C$ simulated from the IFBC1-2 and CONTROL experiments (Figs. 8a-c) present different similar patterns but differ systematically around the ice front regions of the FGL (within ~1 km of the grounding line). Experiments with higher sea levels display smaller $C$ there (Fig. 98, left column) and provide a better match between modeled and simulated surface velocities (Fig. 98, rightmiddle column), which is consistent with the computed RMSD of the surface velocity mismatch (Table 2). If the applied ice front boundary condition underestimates the real world forcing, the inversion process will compensate by increasing the basal dragfriction in this region. However, the large vertical shear strain rate imposes a limit to how much increasing basal drag can reduce the surface velocity, which could explain why the mismatch between the modeled and observed velocity is still large in the narrow band near the ice front (Fig. 9, middle column). For the fast flowing region (velocity > 1500 m a⁻¹), the decreased basal shear force from IFBC2 to IFBC3 (~1.1×10¹¹ N) roughly matches the increased the ice front pressure over a 6 km length of ice front (~2.8×10¹¹ N).

Experiment "IFBC3", with an extreme assumption of applying ice pressure corresponding to a neighbouring column of ice matching the ice front, shows very small basal dragfriction for the ice front area around the grounding line, and also resulted in lower drag over the downstream basin (Fig. 9d8d). 
[revised manuscript text omitted]